# Distinguishing Cause from Effect with Causal Velocity Models

**Johnny Xi** [1]   **Hugh Dance** [2]   **Peter Orbanz** [2]   **Benjamin Bloem-Reddy** [1]

## Abstract

Bivariate structural causal models (SCM) are often used to infer causal direction by examining their goodness-of-fit under restricted model classes. In this paper, we describe a parametrization of bivariate SCMs in terms of a *causal velocity* by viewing the cause variable as time in a dynamical system. The velocity implicitly defines counterfactual curves via the solution of initial value problems where the observation specifies the initial condition. Using tools from measure transport, we obtain a unique correspondence between SCMs and the score function of the generated distribution via its causal velocity. Based on this, we derive an objective function that directly regresses the velocity against the score function, the latter of which can be estimated nonparametrically from observational data. We use this to develop a method for bivariate causal discovery that extends beyond known model classes such as additive or location-scale noise, and that requires no assumptions on the noise distributions. When the score is estimated well, the objective is also useful for detecting model non–identifiability and misspecification. We present positive results in simulation and benchmark experiments where many existing methods fail, and perform ablation studies to examine the method's sensitivity to accurate score estimation.

## 1. Introduction

Distinguishing cause from effect from purely observational data is a challenging task. It is generally impossible to distinguish a causal direction $X \to Y$ from an anti-causal direction $Y \to X$ without intervention, as the corresponding causal graphs are Markov equivalent. To make causal discovery possible, one common approach is to make functional assumptions, for example, non-linear additive noise (Hoyer et al., 2008) or location-scale noise (Xu et al., 2022; Strobl & Lasko, 2023; Immer et al., 2023) on the underlying causal process, and decide on the causal direction based on model fit and/or complexity in both candidate directions.

Likelihood-based approaches seem natural here, as they can be used both for model estimation and evaluation. However, they require the full specification of both the mechanism and the noise distribution. This is undesirable due to the risks of model misspecification, particularly with respect to the noise distribution, which is not directly involved in causality, but can still lead to incorrect causal inference (Schultheiss & Bühlmann, 2023). To avoid that risk, the most prominent example of a functional method is to fit an additive noise model (ANM) using nonparametric regression, which avoids needing to model the noise distribution. Instead of a likelihood-based measure, an independence score (e.g., HSIC, Gretton et al., 2005) between the estimated residuals and cause variable can be used to determine goodness-of-fit (Mooij et al., 2009; Peters et al., 2014). This approach has been extended to location-scale models (LSNM) by Strobl & Lasko (2023); Immer et al. (2023). Separately, Rolland et al. (2022); Montagna et al. (2023b) derived other goodness-of-fit criteria for ANMs that avoid structural model estimation altogether. Instead, those methods extract the signal directly from estimates of the score function $\nabla \log p(x, y)$ under the assumption of Gaussian noise. Subsequent work used score estimation in conjunction with explicit model estimation in the case of non-Gaussian noise (Montagna et al., 2023a). However, we are unaware of any work on functional causal discovery methods beyond the ANM and LSNM classes, which may still be misspecified in many settings.

In this work, we develop a new framework that treats bivariate causal models as dynamical systems, and we use it to devise a new estimation procedure based on the score function of the data, applying it to causal discovery. Specifically, we parametrize a causal model via its *causal velocity*, which describes infinitesimal counterfactuals and can be directly recovered from the score in a simulation-free way analogous to the recent literature on flow-based generative modeling (Lipman et al., 2023; Albergo & Vanden-Eijnden, 2023). Assuming that the marginal and joint score functions can be estimated accurately, our proposed method combines

---

[1]Department of Statistics, University of British Columbia [2]Gatsby Unit, University College London. Correspondence to: Johnny Xi <johnny.xi@stat.ubc.ca>.

*Proceedings of the 42$^{nd}$ International Conference on Machine Learning*, Vancouver, Canada. PMLR 267, 2025. Copyright 2025 by the author(s).

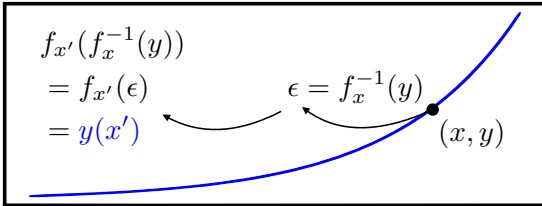

$$v(y, x) = y'(x) = (d/dx')f_{x'}(f_x^{-1}(y))|_{x'=x}$$

*Figure 1.* Given a SCM $X \rightarrow Y$ and an observation $(x, y)$, the causal curve represents the implied counterfactual outcomes had $x$ been $x'$ (top). The derivative of the causal curve, which we call the **causal velocity** function $v(y, x)$, generates all the causal curves and hence characterizes the counterfactuals of the SCM (bottom). In this paper, we show how the velocity can be estimated directly from data using the score function, without the need to evaluate the mechanism itself nor specify the underlying distributions.

the advantages of various state-of-the-art methods in that it is agnostic to the noise distribution and avoids explicit estimation of the functional model, even for non-Gaussian and non-additive cases. Beyond this, the velocity parametrization allows us to specify model classes that extend beyond ANM and LSNM in terms of dependence on the noise, but are still restrictive enough to be useful for causal discovery.

**Contributions** We present a new perspective on invertible SCMs as dynamical systems through their implied velocity functions. In particular, counterfactual prediction becomes the solution to an initial value problem in which the factual observation is the initial condition. This allows us to specify novel causal models. It also allows us to use established ideas from measure transport to connect the causal velocity to the score functions of the data distribution. We use this to devise a novel simulation and likelihood-free estimation procedure for bivariate SCMs and a related goodness-of-fit criterion for bivariate causal discovery.

To support the methodological contributions, we also examine the causal direction identifiability problem from the dynamical perspective, where we obtain a general result that specializes to those existing in the literature. We also establish statistical consistency of our estimation procedure, and show that the rate of convergence is dictated by the rate at which the score estimators converge. We show through synthetic and benchmark experiments that the added model flexibility is beneficial in situations where ANM and LSNM models can fail, but simple parametric velocity families

recover the correct causal direction.[1]

The appealing properties of our method depend on an accurate estimate of the score function at the data. The nonparametric score estimators that we use have known consistency properties and convergence rates (Zhou et al., 2020), which justifies the method asymptotically. However, some distributions, including those seen in common benchmarks, may not have favourable finite-sample behaviour even in the low-dimensional regime. We describe and probe failure modes of our method by way of poor score estimation.

**Outline** In Section 2 we cover the necessary background on bivariate causal discovery and on dynamical systems. Section 3 describes our framework of viewing SCMs as dynamical systems and the velocity parametrization. In Section 4, we use the continuity equation to derive an identity that we use to recover the causal velocity from the score function. Section 5 applies this to causal discovery and discusses consistency and identifiability. Finally, Section 6 and 7 describe related work and experimental evaluations.

## 2. Background

### 2.1. Structural Causal Models

A structural causal model (SCM) for $X$ causing $Y$ is typically defined as the tuple $(f, P_X, P_{\epsilon_y})$,

$$Y = f(X, \epsilon_y), \quad X \perp\!\!\!\perp \epsilon_y, \tag{1}$$

with $X \sim P_X$ and $\epsilon_y \sim P_{\epsilon_y}$. Throughout, we will assume that $f_x(\bullet) := f(x, \bullet)$ is bijective for each $x$, and denote $f_x^{-1}(\bullet)$ as its inverse. The pair $(f_x, P_{\epsilon_y})$ represents the conditional distribution of $Y|X = x$, and $P_X$ completes the specification of the joint distribution. The representation of a conditional distribution as a function of the conditioning variable and independent noise is always possible (Kallenberg, 2021, Lem. 4.22), and it can be chosen to be bijective in the noise if, for each $x$, the conditional distribution of $Y|X = x$ is atomless.

Bijectivity enables identification of counterfactuals. In particular, let $(x, y(x)) = (x, f_x(\epsilon_y))$ denote a realization of the SCM. Assuming no hidden variables, the counterfactual realization "had $x$ been $x'$" is computed by the abduction-action-prediction procedure (Pearl, 2009) as $(x', f_{x'} \circ f_x^{-1}(y(x))) = (x', f_{x'}(x', \epsilon_y)) := (x', y(x'))$. Without bijectivity, the conditional distribution of $\epsilon_y|x, y(x)$ must be estimated. A bijective SCM point-identifies the counterfactual directly as a deterministic transformation:

$$y(x) \mapsto y(x') = f_{x'} \circ f_x^{-1}(y(x)). \tag{2}$$

Nasr-Esfahany et al. (2023) established the identifiability in various settings of bijective SCMs, with $y(x')$ given by

---

[1]Code supporting our experiments can be found on Github at https://github.com/xijohnny/causal-velocity.

*Table 1.* Summary of existing and novel SCMs and their velocities. The velocity perspective allows for specification of novel model classes and can also be used as an alternative means of estimating existing model classes.

| Model | SCM $f(X, \epsilon_y)$ | Counterfactual $f_{x'}(f_x^{-1}(y))$ | Velocity $(d/dx')f_{x'}(f_x^{-1}(y))|_{x'=x}$ | Parameters |
|-------|-----|----------------|----------|------------|
| ANM | $m(X) + \epsilon_y$ | $y + m(x') - m(x)$ | $\dot{m}(x)$ | $\dot{m} : \mathbb{R} \to \mathbb{R}$ |
| PNL | $g(m(X) + \epsilon_y)$ | $g(g^{-1}(y + m(x') - m(x)))$ | $\dot{g}(g^{-1}(y))\dot{m}(x)$ | $\dot{m}, \dot{g} : \mathbb{R} \to \mathbb{R}$ |
| LSNM | $m(X) + e^{h(X)}\epsilon_y$ | $m(x') + e^{h(x')-h(x)}(y - m(x))$ | $\dot{m}(x) + \dot{h}(x)(y - m(x))$ | $\dot{m}, \dot{h} : \mathbb{R} \to \mathbb{R}$ |
| Basis | $\epsilon_y + \int_{x_0}^{x} a^\top \Phi(y(u), u)du$ | $y + \int_x^{x'} a^\top \Phi(y(u), u)du$ | $a^\top \Phi(y, x)$ | $a \in \mathbb{R}^K$ |
| Black box | $\epsilon_y + \int_{x_0}^{x} v(y(u), u)du$ | $y + \int_x^{x'} v(y(u), u)du$ | $v(y, x)$ | $v : \mathbb{R}^2 \to \mathbb{R}$ |

(2). Applying this transformation for all possible $x'$ yields a curve of counterfactual outcomes; see Figure 1.

## 2.2. Functional Bivariate Causal Discovery

Causal discovery aims to determine, from pairs of observed data $(X, Y)$, whether $X$ or $Y$ is the cause. This problem is underdetermined when using observational data, due to the universality of the SCM representation of conditional distributions. One common approach, based on the argument that the data-generating process in the causal direction is simpler (Mooij et al., 2016), is to restrict the model class. It is hoped that in doing so, the model will fit the causal direction but exclude the anti-causal direction (Goudet et al., 2019).

Because the distribution of the noise variables is not related to causality *per se*, most attention is paid to restricting the mechanism. A well-established framework that does not require constraining or modelling the noise distribution is the regression and subsequent independence test (RESIT) approach (Peters et al., 2014), which quantifies the goodness-of-fit of a causal model by the independence of the cause variable to the residuals obtained from the model fit. Originally proposed for nonparametric regression in ANMs, the RESIT approach has also been extended to LSNMs (Immer et al., 2023; Strobl & Lasko, 2023), PNL models by post-processing model estimates with non-linear ICA (Zhang & Hyvärinen, 2009), and rank-based approaches (Keropyan et al., 2023). A key takeaway from this literature is that we should aim for methods that allow for flexible—though not unboundedly so—model classes that can be estimated without specifying the noise distribution.

## 2.3. Flows and Differential Equations

The subsequent sections will expose connections between SCMs and dynamical systems generated by differential equations, so we briefly review some relevant results about the latter here. Our exposition largely follows Arnold (1998, Appendix B) and Santambrogio (2015, Ch. 4.1.2). Consider the ordinary differential equation (ODE) in $\mathbb{R}^d$,

$$\frac{dy(t)}{dt} = v(y(t), t) , \quad y(t) \in \mathbb{R}^d, \ t \in \mathbb{R} . \quad (3)$$

Under regularity conditions, ODEs are known to be in correspondence with *two-parameter flows*, defined as follows.

**Definition 2.1.** A two-parameter flow (or flow) $\varphi_{s,t}$ on $\mathbb{R}^d$ is a continuous mapping

$$(s, t, y) \mapsto \varphi_{s,t}(y), \quad s, t, y \in \mathbb{R} \times \mathbb{R} \times \mathbb{R}^d,$$

that satisfies the flow properties

1. $\varphi_{t,t}(y) = \text{id}(y) = y$, for all $t \in \mathbb{R}, \ y \in \mathbb{R}^d$.

2. $\varphi_{s,t} \circ \varphi_{t,u} = \varphi_{s,u}$ for all $s, t, u \in \mathbb{R}$.

We note that $\varphi_{s,t}^{-1} = \varphi_{t,s}$. The ODE (3) is said to *generate* the flow $\varphi_{s,t}$ if

$$\varphi_{s,t}(y) = y + \int_s^t v(\varphi_{s,u}(y), u) \, du , \quad t \in \mathbb{R} . \quad (4)$$

If $\varphi_{s,t}$ is differentiable in $t$ and satisfies

$$\frac{d}{dt}\varphi_{s,t}(y) = v(\varphi_{s,t}(y), t) \quad \text{and} \quad \varphi_{s,s}(y) = y , \quad (5)$$

then $\varphi_{s,t}$ is said to be a (classical) *solution* of (3). Conversely, a flow can be used to define an ODE. If $t \mapsto \varphi_{s,t}(y)$ is differentiable in $t$ at $t = s$ for all $y$, then

$$v(y, s) := \frac{d}{dt}\varphi_{s,t}(y)\big|_{t=s} \quad (6)$$

yields an ODE to which $\varphi_{s,t}$ is a solution, and $v$ is known as the *velocity*. Under regularity conditions (Appendix A.1), velocities and flows are in one-to-one correspondence.

Now suppose that an initial condition for the ODE (3) is chosen randomly as $Y(s) \sim p_s$, where $p_s$ is the density of a probability distribution on $\mathbb{R}^d$. Letting $Y(s)$ evolve according to the ODE yields $Y(t) = \varphi_{s,t}(Y(s))$. Viewing the flow as a measure transport, the density of $Y(t)$ is $p_t = (\varphi_{s,t})_* p_s$. It can be shown that the family of densities $(p_t)_{t \in \mathbb{R}}$ solves the PDE (known as the *continuity equation*)

$$\partial_t p_t + \nabla_y \cdot (p_t v_t) = 0 , \quad (7)$$

with $v_t := v(\bullet, t)$ and $\partial_t = \partial/\partial t$. It turns out that the solution is unique up to (Lebesgue) almost everywhere equivalence. (See Appendix A.1 for a formal statement.) Note that if $p_t$ is strictly positive then (7) is equivalent to

$$\partial_t \log p_t = -\nabla \cdot v_t - v_t \cdot \nabla_y \log p_t . \quad (8)$$

## 3. SCMs as Dynamical Systems

Generally, flows describe the time evolution of dynamical systems: the usual statement of Definition 2.1 yields the evolution of $y$ between time points, and causality is associated with the forward flow of time. Here, we take a different perspective, viewing counterfactual transformation as a flow in which the role of time is played by a cause variable $x$. Whilst a similar connection has been made in previous work (Dance & Bloem-Reddy, 2024), by specializing to the bivariate real valued case and viewing the cause explicitly as time, we are able to make a direct connection to ODEs. This enables us to derive a novel velocity-based parametrization of SCMs and simulation-free learning procedure. For the remainder of the paper, we work in the setting of $X \in \mathbb{R}$ and $Y \in \mathbb{R}$.

**Definition 3.1.** Let $X$ cause $Y$, with bijective SCM (1). The flow generated by the SCM, or *SCM flow*, is defined as

$$(x, x', y) \mapsto f_{x'}(f_x^{-1}(y)) =: \varphi_{x,x'}(y) . \qquad (9)$$

It is easy to see that Equation (9) satisfies the axioms of a flow Definition 2.1. In particular, for an observation $(x, y)$, the *causal curve* (Figure 1) is the function of $x'$ defined by

$$x' \mapsto y(x') = \varphi_{x,x'}(y) . \qquad (10)$$

Causal curves represent the counterfactual outcomes under the SCM from a factual observation $(x, y)$. Generally, each observation specifies a different causal curve. Based on the dynamical systems perspective reviewed in Section 2.3, $(x, y)$ can be viewed as initial conditions.

In analogy to (6), if $(x, x', y) \mapsto \varphi_{x,x'}(y)$ is continuous and $x \mapsto f_x(\epsilon_y)$ is differentiable for all $\epsilon_y$ then the *causal velocity* of the SCM flow is (see Figure 1, bottom)

$$v(y, x) = \frac{d}{dx'} f_{x'}(f_x^{-1}(y))\big|_{x'=x} . \qquad (11)$$

This holds for all $(x, y)$ in the support of $P_{X,Y}$. The resulting $v$ describes the local behaviour of the flow at $(x, y)$: $v(y, x)$ is the effect of an infinitesimal intervention $x' = x + \delta$ as $\delta \to 0$, which, based on the algebraic properties of the flow, generates the counterfactual curves. Thus, we see that the mechanism of a bijective SCM uniquely determines a flow and its associated velocity.

Conversely, the causal mechanism of a bijective SCM can be parametrized by a velocity $v$ via its corresponding flow from an arbitrary $x_0 \in \mathbb{R}$ acting as the noise state,

$$\varphi_{x_0,x}(y) = y + \int_{x_0}^{x} v(\varphi_{x_0,u}(y), u) \, du . \qquad (12)$$

To complete the specification of the SCM, it remains to specify the conditional distribution $p(y \mid x_0)$ at some arbitrary

point $x_0$, which plays the role of the noise distribution as follows

$$f(X, \epsilon_y) = \varphi_{x_0,X}(\epsilon_y) , \quad \epsilon_y \sim p(y|x_0) .$$

Under regularity conditions, the SCM is unique and holds over all of $\mathbb{R}^2$. Technically, an SCM in the sense of Section 2 also requires $P_X$ to fully specify the joint distribution but it plays no role in the causal mechanism or counterfactuals. The following theorem formalizes the relationship between bijective SCMs and dynamical systems. See Appendix A.1 for technical details on the regularity conditions.

**Theorem 3.2.** *Let $X$ cause $Y$, with $X, Y \in \mathbb{R}$. Under regularity conditions, a bijective SCM uniquely determines a velocity-density pair $(v, p(y|x_0))$. Conversely, $(v, p(y|x_0))$ determines a SCM uniquely up to changes in $P_X$.*

The above equivalence shows that we can view bijective SCMs in terms of their underlying velocity functions without loss of generality. See Table 1 for the velocity functions associated with familiar classes of SCMs.

### 3.1. Velocity Parametrization of SCMs

The dynamical perspective allows a causal model to be specified by the velocity and its implied counterfactuals instead of in terms of first (ANM) or second (LSNM) conditional moments, which can be more interpretable. Viewing $p(y \mid x_0)$ as sampling individual baseline measurements from some population, we see that ANMs indicate that the relative effect of an intervention is the same for all individuals, regardless of their measurement value $y$. This manifests graphically as parallel curves (Figure 2). LSNMs relax this, but implies that the difference between individual trajectories are proportional. Distributionally, ANMs and LSNMs imply that $x$ controls at most the second moment of $p(y \mid x)$. In particular, if $p(y \mid x_0)$ were Gaussian for any $x_0$, then it is also Gaussian for all $x$. On the other hand, the velocity function $e^{y^{-2}}$ implies a more complicated mechanism that can model insensitivity in the tails of $y$, no matter the value of $x$. Distributionally, this also modifies higher order moments of the conditional distribution; see the rightmost panel of Figure 2. In practice, the velocity model can be specified in an interpretable way via domain knowledge of the underlying process, or with basis functions or neural networks for more flexible models. The velocity parametrization automatically specify bijective SCMs, and as such their counterfactuals are identifiable (Nasr-Esfahany et al., 2023).

## 4. Score Functions and SCM Flows

In the previous sections, we established an equivalence between SCMs and an associated pair $(v, p(y|x_0))$. Here, we show that when a joint distribution is generated by an SCM, the velocity leaves an explicit signature on the derivative of

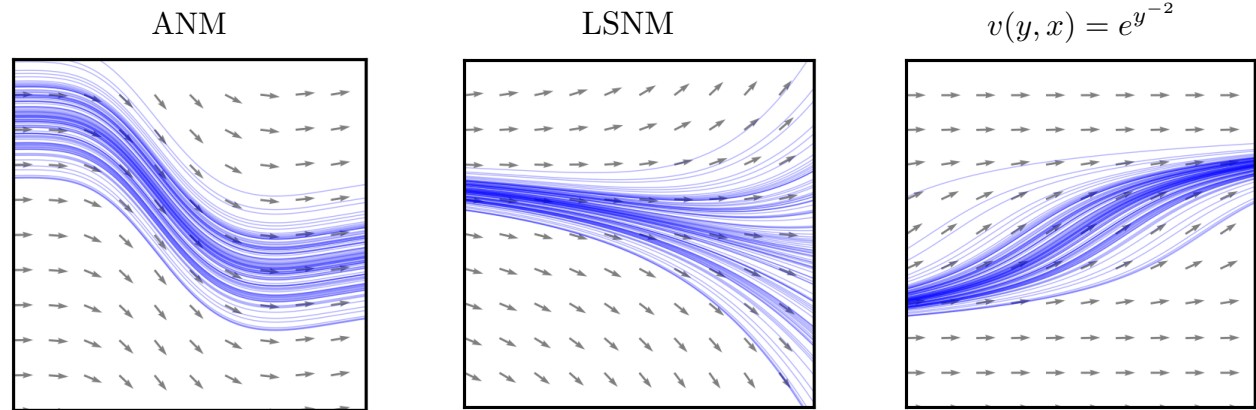

ANM    LSNM    $v(y,x) = e^{y^{-2}}$

*Figure 2.* Distributions of causal curves obtained by sampling 100 initial conditions from $p(y \mid x = x_0) = \mathcal{N}(0,1)$. Note here the ANM and LSNM cross-sections $p(y \mid x)$ are necessarily Gaussian also, no matter how flexible the underlying function classes are. On the other hand, the exponential velocity yields non-Gaussian conditionals at any $x \neq 0$, exhibiting right and left skew as $x < 0$ and $x > 0$, respectively.

the log-density, i.e., the score function. We leverage this observation to derive a goodness-of-fit criterion based on the score, which can then be used for model-fitting and checking directly at the level of the causal velocity. Remarkably, this allows us to check whether data can possibly be generated from an SCM with causal velocity $v$ without ever having to evaluate the mechanism, nor making any assumptions about the underlying distributions beyond differentiable score functions.

Let $v$ be the velocity of a SCM that generates the conditional density $p(y|x)$, and assume that the joint distribution has full support, with differentiable marginal and joint log-densities. Let $s_x(y|x) := \partial_x \log p(y|x)$ and $s_y(y|x) := \partial_y \log p(y|x)$ denote the partial derivatives of the log conditional density, and similarly $s_x(x,y)$ and $s_x(x)$ the partial derivatives of the log joint and marginal densities, respectively. Viewing the cause variable $x$ as time, the $\log$ version of the continuity equation (8) yields

$$s_x(y|x) = -\partial_y v(y,x) - v(y,x)s_y(y|x) . \quad (13)$$

By known results (see Appendix A.1), every solution $p(y|x)$ arises from some initial condition density $p(y|x_0)$ transported by the SCM flow, for fixed but arbitrary $x_0 \in \mathbb{R}$. Moreover, that solution is unique. Therefore, (13) can be used to characterize when a conditional density $p(y|x)$ could possibly have been generated by a SCM with velocity $v$. For practical purposes, (13) can also be stated in terms of marginal and joint scores, which can be estimated from data.

**Theorem 4.1.** *Let $X$ cause $Y$, with $X, Y \in \mathbb{R}$. Assume that $P_{X,Y}$ has full support, with differentiable joint, conditional, and marginal log densities. Then*

$$s_x(x,y) = -\partial_y v(y,x) - v(y,x)s_y(x,y) + s_x(x) \quad (14)$$

*for all $(x,y) \in \mathbb{R}^2$ if and only if $P_{X,Y}$ can be represented by a SCM with velocity $v$ and marginal density $p(x)$.*

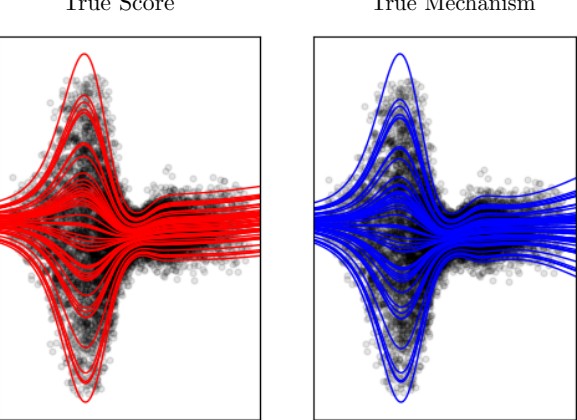

True Score    True Mechanism

*Figure 3.* If the model is well-specified and the true score is given, velocity functions as obtained by minimizing Equation (15) (up to Monte Carlo approximation error) can be integrated numerically (left) to recover the true mechanism (right).

Observe that (14) is entirely in terms of observable scores, and thus is agnostic to the unobserved noise distribution $P_{\epsilon_y}$. Indeed, SCMs with the same mechanism but different noise distributions will still satisfy (14) with their respective scores. This makes it an especially suitable objective for functional causal discovery, which asks whether distributions can be generated by a restricted mechanism class, while leaving the noise distributions unspecified.

## 5. Velocity-Based Causal Discovery

For methods that resolve the direction of causality by fitting a model to the data in both candidate directions, some restriction to model complexity must be made. As discussed in Section 2.2, many existing methods assume some variant of ANM in order to avoid making specific distributional assumptions about the unobserved noise. Our velocity-based

method, described in this section, also makes no assumptions on the noise distribution by capturing its influence via score estimation, and allows all modeling and complexity control to be imposed via the parametrization of the velocity. As established in previous sections, this allows the model to be substantially more flexible than ANMs or LSNMs.

We define a goodness-of-fit (GoF) statistic of a velocity function $v$ as,

$$\mathbf{L}(v) := \mathbb{E}[(s_x(X) - \partial_y v(Y, X) - s_v(X, Y))^2], \quad (15)$$

which is derived from (14) and uses the notation

$$s_v(x, y) := s_x(x, y) + v(y, x) s_y(x, y) . \quad (16)$$

This notation reflects the fact that $s_v$ is the directional derivative of $\log p(x, y)$ along the causal curve, i.e., in the direction $\partial_x(x, y(x)) = (1, v(y(x), x))$ (see Appendix A.2).

We propose to minimize the GoF directly (15), which is zero if and only if (14) holds $P_{X,Y}$-almost everywhere, to estimate the velocity given the scores. See Figure 3 for an example; when the true score is given, minimizing (15) recovers the true mechanism accurately. In practice, the score function is unknown and a two-step approach is required. First, we estimate the score nonparametrically (Zhou et al., 2020); second, we estimate the velocity by minimizing (15), (17) where the scores are replaced by estimators and the expectation is estimated empirically (18).

For causal discovery, we also estimate the model in the $Y \to X$ direction by minimizing

$$\tilde{\mathbf{L}}(\tilde{v}) := \mathbb{E}[(s_y(Y) - \partial_x \tilde{v}(X, Y) - s_{\tilde{v}}(X, Y))^2] . \quad (17)$$

In principle, an estimated velocity could be integrated to a base point in order to estimate the noise variables, i.e., $\hat{\epsilon}_y = \hat{\varphi}_{X,x_0}(Y)$, which could then be used as residuals in independence tests for causal discovery (as in RESIT). However, for the remainder of this paper, we will use the more direct approach of using the value of the objective itself to determine causal direction. The resulting two-step method is as follows:

1. Estimate joint and marginal scores from data.

2. Estimate causal velocity in both directions by minimizing (15), (17), with expectation taken over the data. Choose as causal whichever direction has a lower value of objectives (15), (17) evaluated on the data.

As might be expected, this method requires accurate estimates of the scores. However, since (14) expresses a fixed relationship between the scores and the velocity, an inaccurate score estimate will produce an inaccurate velocity that nonetheless could still yield a small value of (15), thereby potentially introducing error into the subsequent causal discovery. Figure 4 shows that when the true score is given

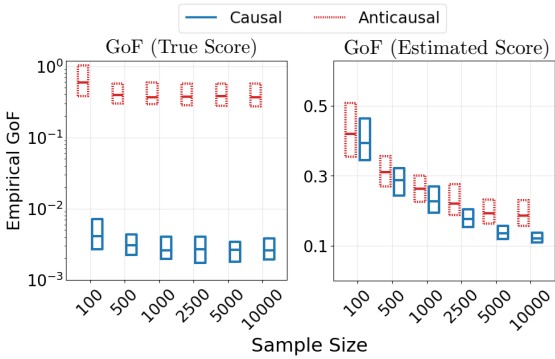

*Figure 4.* If the ground truth score is known, the anticausal GoF is orders of magnitude larger than the causal GoF. In this case, cause can be distinguished from effect with certainty even with as few as 100 samples. Right: As the score estimate quality improves with larger sample sizes, the causal GoF decreases (Theorem 5.1), and the gap between the causal and anticausal GoF widens.

and the model is well-specified, (17) is orders of magnitude larger than (15), which identifies the causal direction with certainty regardless of the sample size. In practice, the causal discovery performance improves as the score estimation improves in sample size (see Section 7.3 for a more in-depth evaluation). In the following section, we show that the empirical estimator of (15) converges at a rate controlled by the rate at which the score estimator converges.

## 5.1. Consistency of Empirical GoF Estimation

Let $(x_i, y_i)_{i=1}^n$ denote the available data. The two-step procedure described above yields the following objective function for estimating the velocity

$$\hat{\mathbf{L}}_n(v) = \frac{1}{n} \sum_{i=1}^{n} ((\hat{s}_x(x_i) - \partial_y v(y_i, x_i) - \hat{s}_v(x_i, y_i))^2 . \quad (18)$$

Recall that $\hat{s}_v(x_i, y_i) = v(y_i, x_i) \hat{s}_y(x_i, y_i) + \hat{s}_x(x_i, y_i)$, and here $\hat{s}_x, \hat{s}_y$ are the score estimators, for which we use either the Stein estimator (Li & Turner, 2018) or one based on a kernel density estimator (Wibisono et al., 2024). These estimators can both be viewed as doing regularized, vector-valued reproducing kernel Hilbert space (RKHS) regression on the true scores (Zhou et al., 2020). Based on this, the following result shows that our estimator for the loss function converges at a rate no worse than the convergence rate of the score function estimators. In the following theorem we denote $s_{x,y}^{(x)} := s_x(x, y)$, $s_{x,y}^{(y)} := s_y(x, y)$ for convenience, and similarly for the corresponding estimators.

**Theorem 5.1.** *Let $s_x \in \mathcal{H}_\mathcal{X}$ and $s_{x,y}^{(x)}, s_{x,y}^{(y)} \in \mathcal{H}_{\mathcal{X},\mathcal{Y}}$ where $\mathcal{H}_\mathcal{X}, \mathcal{H}_{\mathcal{X},\mathcal{Y}}$ are RKHSs induced by bounded kernels $k_x < B, k_{x,y} = k_x \otimes k_y < B$. Additionally, let $v \in C_b^1(\mathbb{R}^2, \mathbb{R})$. If the score estimators have bounded RKHS norm that converges, such that each of $\|\hat{s}_x - s_x\|_{\mathcal{H}_\mathcal{X}}$, $\|\hat{s}_{x,y}^{(x)} - s_{x,y}^{(x)}\|_{\mathcal{H}_{\mathcal{X},\mathcal{Y}}}$,*

$\|\hat{s}_{x,y}^{(y)} - s_{x,y}^{(y)}\|_{\mathcal{H}_{\mathcal{X},\mathcal{Y}}}$ *is* $\mathcal{O}_p(n^{-\frac{1}{\alpha}})$ *for some* $\alpha > 0$, *then* $\hat{\mathbf{L}}_n(v) - \mathbf{L}(v) = \mathcal{O}_p(n^{-\frac{1}{\beta}})$, *for* $\beta = \max\{\alpha, 2\}$.

Zhou et al. (2020) showed that under smoothness assumptions on each of $\log p(x, y), k_x, k_y$, then convergence at rate $\alpha \in [3, 4]$ can be achieved. See Figure 4 (right) for an empirical demonstration of the behaviour of $\hat{\mathbf{L}}_n(v)$ in a well-specified model, and Section 7.3 for the implications on causal discovery performance.

### 5.2. Identifiability

In the limit of infinite data, the causal direction of an ANM is known to be identifiable under certain conditions, formulated as the non-existence of a solution to a differential equation for $\log p(x)$ (Hoyer et al., 2008; Zhang & Hyvärinen, 2009). For fixed ANM parameters $m, p_\epsilon$, the set of marginal densities that satisfy the differential equation (and hence are non-identifiable) is a three-dimensional space contained in an infinite-dimensional space, and therefore it is believed that ANMs can identify causal direction in "most cases" (Peters et al., 2014). Identifiability in ANMs is the strongest known result, as the characterization does not depend simultaneously on the parameters of both model directions, and therefore holds uniformly over the model space.

As argued by Tagasovska et al. (2020); Xu et al. (2022); Strobl & Lasko (2023); Immer et al. (2023) and others, additive noise can be an overly restrictive assumption: even basic heteroscedastic noise (i.e., a LSNM) can lead ANM-based causal discovery procedures to mis-identify the correct causal direction. While LSNMs extend ANMs, even that straightforward generalization significantly complicates the identifiability analysis. In analogy to ANM identifiability, Strobl & Lasko (2023); Immer et al. (2023) derived a partial differential equation (PDE) whose solutions characterize non-identifiability. In contrast to ANMs, the LSNM PDE depends simultaneously on the parameters of both model directions. Therefore, it is useful on a case-by-case basis, but does not yield a characterization of non-identifiability uniformly over the model class. As we show in Appendix C, a similar characterization for velocity models can be obtained by analyzing the continuity equation (14) in both directions.

**Proposition 5.2.** *Assume that the joint distribution of the observed data can be expressed as a causal velocity model in both directions,*

$$Y \overset{\mathrm{d}}{=} \varphi_{x_0, X}(\epsilon_y), \quad X \perp\!\!\!\perp \epsilon_y,$$
$$X \overset{\mathrm{d}}{=} \tilde{\varphi}_{y_0, Y}(\epsilon_x), \quad Y \perp\!\!\!\perp \epsilon_x,$$

*with corresponding velocities $v$ and $\tilde{v}$, respectively. Then, as long as $\pi(x, y) = \log p(x, y)$ and the required derivatives exist, the velocities satisfy on the support of $p(x, y)$,*

$$\partial_y^2 v + \partial_y(v \cdot \pi) = \partial_x^2 \tilde{v} + \partial_x(\tilde{v} \cdot \pi).$$

Expressions for the relevant partial derivatives of $\pi$ in terms of the forward model parameters $\varphi, v$ are given in Appendix C. We show in Appendix C.1 how the known ANM non-identifiability differential equation follows naturally from this characterization. In Appendix C.2, we use it to derive a criterion for non-identifiability that holds uniformly over the class of LSNMs, albeit at a loss of interpretability. Thus, we expect that a general analysis of identifiability—if one is possible—may require new techniques.

## 6. Related Work

**Score Matching ANMs** A recent line of work also use signatures in the score function for causal discovery in ANMs. (Rolland et al., 2022; Montagna et al., 2023b) use properties of Gaussian noise to derive conditions on the score that are satisfied by non-cause variables. (Montagna et al., 2023a) show that even under arbitrary distributions the score is a deterministic function of the noise variables, and fit the model using nonparametric regression to evaluate this condition. These methods are restricted to the ANM case, but are focused on sink node (non-cause) identification in multivariate SCMs, while our focus is on more general model classes in the bivariate setting. The continuity equation (13) simplifies for ANMs, and we show how to recover the identities used in this line of work in Appendix A.3.

**Cocycles in Causal Modeling** Connections between counterfactual causal models and dynamical systems have been previously established by Dance & Bloem-Reddy (2024) in a more general setting, using a class of maps called cocycles. A two-parameter flow is an example of a cocycle. Dance & Bloem-Reddy (2024) focused on distribution-robust inference with a known causal ordering with multiple cause variables, rather than bivariate discovery. Furthermore, they do not analyze the cocycle as a dynamical system nor make any connection to the score function.

**Continuous Normalizing Flows** A popular class of velocity-based probabilistic model are continuous normalizing flows (CNF) (Chen et al., 2018). There, instead of conditioning, time is an auxiliary variable, and the generative model is for a marginal density $\varphi_{0,1}(y_0)$, where $y_0$ is drawn from an arbitrary base distribution. Our learning objective (18), which targets conditional distributions instead, is similar to recently proposed simulation-free objectives, which attempt to target the velocity directly in CNFs, e.g., Flow Matching (Lipman et al., 2023). Normalizing flows have been used for causal inference (Khemakhem et al., 2021; Javaloy et al., 2024) with a known causal graph. The causal auto-regressive flow model (Khemakhem et al., 2021) when restricted to the bivariate case represents a flexible LSNM, and was used for likelihood-based causal discovery. Tu et al. (2022) propose to analyze the CNF velocity as an alternative decision rule for causal discovery in ANMs.

*Table 2.* Accuracy (and AUDRC) of velocity models for determining the correct causal direction, evaluated on a collection of synthetic datasets (columns). Score is estimated using the Stein method with the Gaussian kernel. The results shown for LOCI (NN/Spline), IGCI, and CGCI represent the best performance over different variants. The best performing velocity model, and, if applicable, the overall best model, are indicated in **bold**. KDE results and a subsampled $n = 1000$ setting can be found in Appendix E.1.

| Model | Velocity | Sigmoid | ANM | LSNM |
|---|---|---|---|---|
| B-LIN | 91 (97) | **87 (95)** | 66 (70) | 66 (77) |
| B-QUAD | **97 (98)** | 80 (94) | 63 (70) | 73 (79) |
| V-ANM | 86 (97) | 36 (41) | **92 (95)** | 59 (67) |
| V-LSNM | 89 (99) | 81 (95) | 86 (92) | **82 (92)** |
| V-NN | 96 (99) | 56 (69) | 65 (72) | 57 (64) |
| LOCI (HSIC) | 40 (25) | 70 (86) | **97 (99)** | 83 (89) |
| LOCI (Lik) | 43 (66) | 43 (63) | 49 (56) | 40 (31) |
| CDS | 56 (47) | 12 (5) | 92 (99) | 59 (68) |
| IGCI | 66 (77) | 73 (83) | 29 (21) | 37 (31) |
| RECI | 33 (26) | 13 (4) | 34 (36) | 38 (55) |
| CGCI | 52 (51) | 41 (33) | 85 (96) | 76 (89) |

# 7. Experiments

We evaluate our method empirically on new synthetic datasets as well as on existing benchmarks from the literature (Mooij et al., 2016; Immer et al., 2023). We study the relative performance of different velocity model classes, including velocity-based parametrizations of known classes. In Section 7.3, we show empirically that the performance of our method hinges on the accuracy of the score estimators. When the true score is given, our method achieves perfect causal discovery with as few as $n = 100$ samples (Figure 4).

**Velocity Estimation** We use three novel parameterizations of the velocity: a 2-layer MLP (V-NN), and linear/quadratic basis families (B-LIN/B-QUAD). We also experiment by fitting ANM and LSNMs via their velocity parameterizations (V-ANM and V-LSNM). See Table 1 and Appendix D for details. Given an estimate of the score, we minimize (18) using the Adam optimizer (Kingma & Ba, 2015) to estimate the velocity. The term $\partial_y v(y, x)$ is obtained by automatic differentiation. The datasets we consider are small ($n < 10^5$), and thus we take full batch gradient steps. To mitigate potential non-identifiability and regularize estimation, we penalize the complexity of the model via higher order derivative terms $d^k y(x)/dx^k$ (Kelly et al., 2020) in the case where both directions are specified to encourage selecting the simpler model. In practice we use $k = 2$.

**Score Estimation** For estimation of the score, we use the Stein gradient estimator of Li & Turner (2018) (STEIN), with the Gaussian kernel and bandwidth selected by the median heuristic. We also consider the derivative of the log of a standard kernel density estimator with empirical Bayes smoothing (Wibisono et al., 2024) (KDE). For the KDE-based estimator, we use the exponential (Laplace) kernel,

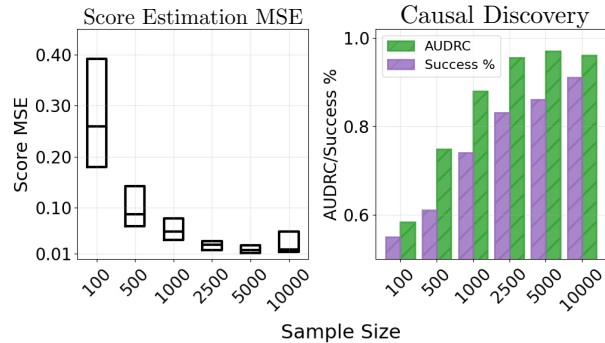

*Figure 5.* Score estimation and causal discovery performance as sample size increases in a well-specified ANM benchmark. Upper and lower limits indicate 1st–3rd quartile over 100 datasets. The larger error at $n = 10000$ is likely due to an increased chance of encountering outliers; note the median continues to decrease.

which we found to result in the best performance.

**Computation** Experiments are written in JAX (Bradbury et al., 2018) and carried out either on a M1 Mac or NVIDIA V100 GPU. The most costly operation is the Stein score estimation, which is $O(n^3)$ due to the inversion of an $n \times n$ matrix (Li & Turner, 2018). Note this only has to be performed once, prior to training the velocity model. With $n = 1000$ and $n = 5000$, the median time is ~0.02s and ~0.3s respectively on GPU, and ~0.9s and ~15s on M1.

**Evaluation** Causal direction is determined by comparing the empirical loss (18) in both directions and selecting the smaller value. Either the squared or absolute values can be used; here we use the squared value. We follow the literature and report the raw causal discovery accuracy as well as the associated area under the decision rate curve (AUDRC).

**Other Methods** We compare our method to LOCI, an LSNM-based model which represents the state-of-the-art in functional causal discovery based on the results reported by Immer et al. (2023). It has two major variants, one based on the Gaussian conditional likelihood (Lik) which makes strong parametric assumptions on the conditional distribution, and one based on the RESIT approach (HSIC), which is robust to the noise distribution and thus more comparable to our method. We also compare our method to non-functional methods that do not make parametric assumptions such as CDCI (Duong & Nguyen, 2022) and other methods that are implemented in the Causal Discovery Toolbox (Kalainathan et al., 2020), which include CDS (Fonollosa, 2019), IGCI (Daniušis et al., 2010), and RECI (Blöbaum et al., 2018).

## 7.1. Datasets

**Synthetic** We design two synthetic settings for which existing functional methods (e.g., LOCI) are misspecified and are expected to fail. All benchmarks consist of 100 datasets of size $n = 5000$ with randomly sampled mechanisms. The

*Table 3.* Accuracy (and AUDRC) of velocity models on benchmark data. LOCI (NN/Spline), IGCI, and CGCI represent best results over all variants. The best performing velocity model, and, if applicable, the overall best model, are indicated in **bold**. Additional comparisons on these benchmarks can be found in Mooij et al. (2016); Immer et al. (2023).

| Model | SIM | | SIM-C | | SIM-G | | Tü | | Tü (cts.) | |
|---|---|---|---|---|---|---|---|---|---|---|
| | KDE | STEIN | KDE | STEIN | KDE | STEIN | KDE | STEIN | KDE | STEIN |
| B-LIN | 40 (39) | 58 (62) | 56 (50) | 55 (54) | 83 (93) | 71 (85) | **71 (76)** | 54 (59) | **78 (81)** | 55 (68) |
| B-QUAD | 39 (37) | 60 (64) | 43 (37) | 62 (54) | **94 (99)** | 71 (82) | 52 (65) | 56 (58) | 58 (68) | 61 (68) |
| V-ANM | 36 (39) | **63 (77)** | 52 (49) | **76 (71)** | 58 (77) | 77 (93) | 68 (78) | 60 (64) | 74 (82) | 69 (72) |
| V-LSNM | 39 (39) | 62 (67) | 52 (49) | **76 (71)** | 74 (92) | 72 (88) | 58 (70) | 62 (65) | 57 (73) | 68 (73) |
| V-NN | 37 (38) | 57 (69) | 48 (44) | 60 (54) | 84 (95) | 68 (78) | 58 (70) | 59 (65) | 65 (72) | 64 (68) |
| LOCI (HSIC) | **79 (89)** | | 83 (93) | | 81 (91) | | 60 (56) | | 57 (58) | |
| LOCI (Lik) | 52 (68) | | 50 (63) | | 78 (89) | | 57 (66) | | 60 (66) | |
| CDS | 68 (85) | | 78 (89) | | 73 (82) | | 66 (62) | | 60 (55) | |
| IGCI | 37 (33) | | 42 (35) | | 86 (96) | | 60 (71) | | 53 (64) | |
| RECI | 45 (48) | | 56 (58) | | 42 (41) | | **72 (88)** | | 71 (88) | |
| CGCI | 68 (84) | | 76 (91) | | 75 (88) | | 67 (64) | | 61 (57) | |

Velocity benchmark is generated by numerically integrating periodic velocity functions with initial values given by the noise variable, as in (12). The Sigmoid benchmark can be seen as a variation on LSNMs with additional post-nonlinear and affine transformations.

In all cases, noise variables are sampled as a randomly sampled invertible transformation (i.e., a random CDF transform) of Gaussian noise. As such, methods based on Gaussianity are also expected to fail. To illustrate our method as an alternative to HSIC in existing functional classes, we also design synthetic ANM and LSNM benchmarks for which V-ANM and V-LSNM are well-specified. Sample plots of datasets are given in Appendix D.1.

**Benchmark Data** Following previous work (Blöbaum et al., 2018; Tagasovska et al., 2020; Immer et al., 2023), we also evaluate our method on the SIM-series of simulated benchmarks and the Tübingen Cause-Effect pairs of Mooij et al. (2016). Since our method requires the existence of log-densities, we also test removing instances from the Tübingen collection with integer-valued observations, which we refer to as **Tü (cts.)** (details in Appendix D.4).

## 7.2. Results

The experiments show that our two-step method, in particular with B-LIN and B-QUAD velocity models which represents novel model classes, are able to distinguish cause from effect when existing methods fail (Table 2). For non-Gaussian ANM and LSNM data, velocity parametrizations achieve performance that is competitive with the HSIC variant of LOCI. As expected, the (Gaussian) likelihood variant of LOCI performs poorly in our non-Gaussian simulations. Our method also achieves state-of-the-art performance on certain benchmarks, in particular the Gaussian SIM-G dataset which we suspect is due to the score being

well-estimated, and is competitive with other methods overall (Table 3). Notably, taking only the continuous subset of the Tübingen dataset improves performance in our method but not existing methods, which emphasizes the importance of the log-density assumption. Interestingly, using the KDE estimate improves causal discovery performance in cases where the noise distributions are Gaussian (SIM-G), but also on the real data benchmark (Tü).

## 7.3. Sample Size and Score Estimation

To study the effect of score estimation on velocity estimation and downstream causal discovery, we also designed synthetic ANM and LSNM datasets with Gaussian noise, for which the true marginal cause and joint scores can be obtained analytically. For the effect variable, we compute the marginal density by Monte Carlo integration of the conditional and differentiate to obtain the marginal score.

In Figure 4, we saw that using the ground truth scores as an input to estimate the velocity determined the causal direction with certainty, regardless of the sample size used in minimizing (18). On the other hand, when the score is estimated, causal discovery performance appears to improve as sample sizes grows. Here, we provide additional evidence that the improvement in causal discovery performance is due to an improving score estimate. We use the ground truth scores to evaluate the quality of our estimated scores by evaluating the mean squared estimation error. Figure 5 shows that the score estimation largely improves in sample size, and this corresponds to an improvement in both measures of causal discovery performance. Note the figure shows estimation of the marginal score of the effect variable—joint and marginal cause score estimation follows similar patterns. See Appendix E.3 for full tables of results. We believe improvements to score estimation will directly benefit the applicability of our methodology in future practice.

## Acknowledgments

The authors are grateful to the anonymous reviewers, whose comments helped improve the paper. This research was supported in part through computational resources and services provided by Advanced Research Computing at the University of British Columbia. JX is supported by an Natural Sciences and Engineering Research Council of Canada (NSERC) Canada Graduate Scholarship. BBR acknowledges the support of NSERC: RGPIN2020-04995, RGPAS-2020-00095. HD and PO are supported by the Gatsby Charitable Foundation.

## Impact Statement

This paper presents work whose goal is to advance the field of Machine Learning. There are many potential societal consequences of our work, none which we feel must be specifically highlighted here.

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

# A. Additional Details

## A.1. Background on Flows

The key results we use from dynamical systems are as the following two theorems. We refer the reader to Arnold (1998); Santambrogio (2015), and references therein, for more details.

**Theorem A.1** (Arnold, 1998, Thm. B.3.1, B.3.5). *Suppose that $(t, y) \mapsto v(y, t)$ is locally Lipschitz continuous in $y$, and satisfies the local linear growth condition,*

$$|v(y, t)| \leq a(t)|y| + b(t) ,$$

*where $a$ and $b$ are non-negative locally integrable functions. Then the maximal solution to* (3) *generates a unique solution flow $\varphi_{s,t}$ as in Equations* (4) *and* (5). *If $v(\bullet, t)$ is $k$-times continuously differentiable in $y$ for each $t$, then so is $\varphi_{s,t}(\bullet)$. Conversely, if a flow $\varphi_{s,t}(y)$ is differentiable in $t$ at $t = s$ for every $y$, then the ODE defined by* (6) *both generates and is solved by $\varphi_{s,t}$.*

Flows generated by velocities also characterize the possible solutions to the continuity equation.

**Theorem A.2** (Santambrogio, 2015, Thm. 4.4). *Fix $v$ and assume the conditions of Theorem A.1. Then for any $p_s$, the family of densities $p_t = (\varphi_{s,t})_* p_s$ uniquely solves the continuity equation* (7) *with initial condition density $p_s$. Moreover, every solution $(p_t)_{t \in \mathbb{R}}$ to* (7) *for fixed $v$ is obtained from the corresponding flow of some initial condition density $p_s$.*

## A.2. Density Along a Causal Curve

The continuity equation in (14) can be viewed as an Eulerian perspective on the problem, tracking velocity and density at points $(x, y)$. Alternatively, we may consider the Lagrangian perspective, viewing the problem along trajectories, which in this case are causal curves. The result is the same, but we include the analysis here to shed additional light on the problem.

Let $y(x)$ be a causal curve with velocity $v$. We evaluate the log joint density along the curve $(x, y(x))$:

$$\log p(x, y(x)) = \log p(x) \tag{19}$$
$$+ \log p(y(x_0) \mid x_0) + \log \left| \frac{\partial \varphi_{x, x_0}(y(x))}{\partial y(x)} \right| ,$$

where $x_0$ is an arbitrary origin point. This follows from the usual change-of-variables formula that results from transporting along the causal curve from $(x, y(x))$ to $(x_0, y(x_0))$. Compared to the usual SCM construction, $\log p(y(x_0) \mid x_0)$ is the log-density of the noise, and $\varphi_{x, x_0}$ represents the residual map. Notice this "noise" term does not depend on $x$. This is because the noise realization is the same no matter where the counterfactual prediction is made.

Now, we take a (total) derivative in $x$, and use the instantaneous change-of-variables formula commonly used in the neural ODEs literature (Chen et al., 2018; Hodgkinson et al., 2021) to obtain

$$\frac{d \log p(x, y(x))}{dx} = \frac{d \log p(x)}{dx} - \frac{\partial v(y(x), x)}{\partial y(x)}$$
$$= s_x(x) - \frac{\partial v(y(x), x)}{\partial y(x)}. \tag{20}$$

Notice the first term is precisely the marginal score $s_x(x)$. We can take the same total derivative using the chain rule as

$$\frac{d \log p(x, y(x))}{dx}$$
$$= \frac{\partial \log p(x, y(x))}{\partial x} + \frac{dy(x)}{dx} \frac{\partial \log p(x, y(x))}{\partial y(x)}$$
$$= s_x(x, y(x)) + v(y(x), x) s_y(x, y(x)) \tag{21}$$

In fact, this is a directional derivative of the joint log-density along the causal curve, i.e., in the direction $\partial_x(x, y(x)) = (1, v(y(x), x))$. Hence, denoting

$$s_v(x, y(x)) := s_x(x, y(x)) + v(y(x), x) s_y(x, y(x)) , \tag{22}$$

and equating (20) and (21), we obtain an expression equivalent to (14),

$$s_x(x) - \partial_{y(x)} v(y(x), x) = s_v(x, y(x)). \tag{23}$$

This is a relation between the marginal and joint log-density functions that is satisfied when the conditional distribution arises from an SCM with velocity $v$. It is intuitive that this is in terms of $v$ and the score, which both characterize local change.

### A.3. Score-based ANM Methods

Rolland et al. (2022); Montagna et al. (2023a;b) also use the score function for functional causal discovery. There, the focus is on finding leaf nodes in a multivariate causal graph, that is, non-cause nodes. This is equivalent to finding the effect variable in the bivariate setting. Here, we show how the conditions derived there can be interpreted via the continuity equation. Throughout, let $y$ be the effect variable so that $X \to Y$ is the causal graph.

The continuity equation in our setting states (7)

$$s_x(y|x) = -\dot{m}(x) s_y(y|x) , \tag{24}$$

where note for an ANM, $v(y, x) = \dot{m}(x)$ and hence $\partial_y v(y, x) = 0$. Recall that for an ANM we also have

$$p(y \mid x) = p_\epsilon(y - m(x)), \tag{25}$$

where $p_\epsilon$ is the density of the noise variable $\epsilon_y$. Let $s_\epsilon = \partial_\epsilon \log p_\epsilon(\epsilon)$, then the continuity equation becomes

$$-\dot{m}(x) s_\epsilon(\epsilon) = -\dot{m}(x) s_y(y \mid x), \tag{26}$$

where we wrote $\epsilon = y - m(x)$ on the LHS. Now, noting that $s_y(y \mid x) = s_y(x, y)$, the $y$ component of the joint score, we have

$$s_\epsilon(\epsilon) = s_y(x, y), \tag{27}$$

which holds for all $(x, y)$ where $\dot{m}(x) \neq 0$ (thus, for all $x$ if $m$ is injective, corresponding to Condition (2d) in Appendix B.1). The above expression is precisely the general one used by Montagna et al. (2023a) for general non-Gaussian noise. There, they use score estimation to estimate $s_y(x, y)$, then fit a non-parametric regression model to estimate $m$ and obtain the residuals $\hat{\epsilon}$. Then, the equation above says that for the effect variable, it is possible to perfectly predict $s_y(x, y)$ from $\hat{\epsilon}$.

Under the Gaussian noise assumption $\epsilon_y \sim \mathcal{N}(0, \sigma^2)$, Rolland et al. (2022); Montagna et al. (2023b) derive a more specific equation that is easily derived independently of the continuity equation. Consider the $y$ component of the joint score,

$$s_y(x, y) = s_y(y \mid x) = s_\epsilon(y - m(x)). \tag{28}$$

Under the Gaussian noise assumption, $s_\epsilon(\epsilon) = -\epsilon/\sigma^2$. Thus, we have

$$s_y(x, y) = \frac{m(x) - y}{\sigma^2}. \tag{29}$$

This indicates that $\partial_y s_y(x, y) = -\sigma^{-2}$, which is constant over $(x, y)$. Rolland et al. (2022) hence devise an algorithm to estimate the Hessian of the log-likelihood (i.e., Jacobian of the score) and select the node with minimum empirical variance as the effect.

## B. Proofs

### B.1. Regularity Conditions of SCM-Velocity Correspondence

The correspondence in Theorem 3.2 between SCMs and velocity-initial condition density pairs $(v, p(y|x_0))$ essentially follows from the known results on dynamical systems reviewed in Appendix A.1. The only subtleties are under what conditions the relationships hold over all of $\mathbb{R}^2$, rather than on a subset. We discuss them here before proving Theorem 3.2.

1. $P_{X,Y}$ has full support on $\mathbb{R}^2$ and is absolutely continuous with respect to Lebesgue measure.

2. For any SCM mechanism $f$:

    (a) For each $x \in \mathbb{R}$, $f(x, \bullet)$ is a bijection $\mathbb{R} \to \mathbb{R}$.

    (b) The mapping $(x, x', y) \mapsto f_{x'}(f_x^{-1}(y))$ is continuous for each $x, x', y \in \mathbb{R}^3$.

    (c) The mapping $x' \mapsto f_{x'}(f_x^{-1}(y))$ is differentiable at $x' = x$ for all $x, y \in \mathbb{R}^2$.

    (d) There is a unique $x_0 \in \mathbb{R}$ such that for all $\epsilon_y \in \mathbb{R}$, $f(x_0, \epsilon_y) = \epsilon_y$.

3. For any velocity field $v$:

    (a) For each $x \in \mathbb{R}$, $v(\bullet, x)$ is locally Lipschitz continuous or $k$-times continuously differentiable.

    (b) Satisfies

$$|v(y, x)| \le a(x)|y| + b(x) \,,$$

where $a$ and $b$ are non-negative locally integrable (integrable on every compact subset of $\mathbb{R}$) functions.

Within the confines of standard practice, these assumptions are not restrictive. Assumption 2(d) may require some care, but is easy to achieve in practice. For example, with LSNMs, it becomes

$$m(x_0) + e^{h(x_0)}\epsilon_y = \epsilon_y \,,$$

which requires each of $m, h$ to have a unique zero at $x_0$. Assuming that each of $m, h$ have at least one zero, if they are injective then the zero is unique.

In general, assuming that at least one such $x_0$ exists, a sufficient condition for uniqueness is that $x \ne x'$ implies that $f(x, \epsilon_y) \ne f(x', \epsilon_y)$ for *some* $\epsilon_y \in \mathbb{R}$.

Assumption 3(b) on the velocity ensures that a local solution to the ODE $dy/dx = v(y, x)$ extends to a unique global solution

$$\varphi_{x_0,x}(y) = y + \int_{x_0}^{x} v(\varphi_{x_0,u}(y), u) \, du \,,$$

whereas assumption 3(a) ensures that

$$\frac{d}{dx}\varphi_{x_0,x}(y) = v(\varphi_{x_0,x}(y), x) \,, \quad \varphi_{x_0,x_0}(y) = y \,,$$

holds for all $x$ in the solution domain. See, for example, Arnold (1998, Appendix B) for details.

*Proof of Theorem 3.2.* First, assume that $P_{X,Y}$ has full support on $\mathbb{R}^2$ and is specified by a bijective SCM

$$Y = f(X, \epsilon_y) \,, \quad X \perp\!\!\!\perp \epsilon_y$$

with noise density $\epsilon_y \sim p_0$. Then

$$\varphi_{x,x'}(y) = f_{x'}(f_x^{-1}(y))$$

defines a continuous flow. Along with the assumed differentiability in 2(c), by (Arnold, 1998, Thm. B.3.5),

$$v(y, x) := \left. \frac{d}{dx'} f_{x'}(f_x^{-1}(y)) \right|_{x'=x}$$

defines the velocity that generates the flow. As long as the mechanism is bijective on all of $\mathbb{R}$ for each $x$ then this relationship holds over all of $\mathbb{R}^2$. By the uniqueness of $x_0$, $p(y|x_0) = p_0(y)$ is uniquely specified by the SCM.

Conversely, fix a pair $(v, p(y|x_0))$ such that $p(y|x_0)$ has full support on $y \in \mathbb{R}$. As long as $v$ is sufficiently regular so as to yield a flow $(x, y) \mapsto \varphi_{x_0,x}(y)$ over all of $\mathbb{R}^2$, then the flow is unique and therefore

$$f(X, \epsilon_y) := \varphi_{x_0,X}(\epsilon_y) \,,$$

with $\epsilon_y \sim p(y|x_0)$ uniquely specifies a bijective SCM, except for the marginal distribution $P_X$. Assumption 3(b) above guarantees the global uniqueness of the flow-based mechanism. $\square$

## B.2. Proof of Theorem 4.1

*Proof of Theorem 4.1.* By Theorem A.2, the continuity equation is uniquely solved by densities generated by the flow associated with $v$ and some initial condition density $p(y|x_0)$. Hence, if $P_{X,Y}$ can be represented by a SCM with velocity $v$ then its conditional $p(y|x)$ will satisfy the continuity equation

$$\partial_x p(y|x) = -\partial_y (p(y|x) \cdot v(y,x)) .$$

Since $P_{X,Y}$ (and hence $P_{Y|X}$) is assumed to have full support, its density will be strictly positive and the continuity equation can be written

$$\frac{\partial_x p(y|x)}{p(y|x)} = -\partial_y v(y,x) - v(y,x) \frac{\partial_y p(y|x)}{p(y|x)}$$
$$\partial_x \log p(y|x) = -\partial_y v(y,x) - v(y,x) \partial_y \log p(y|x) .$$

Noting that $\partial_y p(y|x) = \partial_y p(x,y)$ and adding $\partial_x \log p(x)$ to both sides, we have

$$\partial_x \log p(x,y) = -\partial_y v(y,x) - v(y,x) \partial_y \log p(x,y) + \partial_x \log p(x) ,$$

which is (14).

Conversely, if $P_{X,Y}$ satisfies (14) then $p(y|x)$ must satisfy the continuity equation and therefore the associated velocity can be used to represent $p(y|x)$ with the SCM constructed as in Theorem 3.2. $\qquad\square$

## B.3. Proofs for Section 5.1

**Theorem B.1.** *Let $s_x \in \mathcal{H}_{\mathcal{X}}$ and $s_{x,y}^{(x)}, s_{x,y}^{(y)} \in \mathcal{H}_{\mathcal{X},\mathcal{Y}}$ where $\mathcal{H}_{\mathcal{X}}, \mathcal{H}_{\mathcal{X},\mathcal{Y}}$ are RKHSs induced by bounded kernels $k_x < B$, $k_{x,y} = k_x \otimes k_y < B$. Additionally, let $v \in C_b^1(\mathbb{R}^2, \mathbb{R})$. Then, if the score estimators have bounded norm and converge in RKHS norm, such that each of $\|\hat{s}_x - s_x\|_{\mathcal{H}_{\mathcal{X}}}$, $\left\|\hat{s}_{x,y}^{(x)} - s_{x,y}^{(x)}\right\|_{\mathcal{H}_{\mathcal{X},\mathcal{Y}}}$, $\left\|\hat{s}_{x,y}^{(y)} - s_{x,y}^{(y)}\right\|_{\mathcal{H}_{\mathcal{X},\mathcal{Y}}}$ is $\mathcal{O}_p(n^{-\frac{1}{\alpha}})$ for some $\alpha > 0$, then $\hat{\mathbf{L}}_n(v) - \mathbf{L}(v) = \mathcal{O}_p(n^{-\frac{1}{\beta}})$, for $\beta = \max\{\alpha, 2\}$.*

*Proof.* For convenience in this proof, we will use the notation $s_1 := s_x$, $s_2 := -s_{x,y}^{(x)}$, $s_3 := -s_{x,y}^{(y)}$, $h_1 := -\frac{\partial v}{\partial y}$, $h_2 = v$, and $Z = (X,Y) \sim p_{x,y}$. Note that by assumption, we have that $s = (s_1, s_2, s_3) \in \mathcal{H}$, where $\mathcal{H} = \mathcal{H}_1 \otimes \mathcal{H}_2 \otimes \mathcal{H}_3$ is a tensor product reproducing kernel Hilbert space with components $\mathcal{H}_1 := \mathcal{H}_{\mathcal{X}}$, $\mathcal{H}_2 = \mathcal{H}_3 := \mathcal{H}_{\mathcal{X},\mathcal{Y}}$. Using this notation, we can express Equation (15) and its estimator Equation (18) as

$$\mathbf{L}(v) = \mathbb{E} f(s, Z) \tag{30}$$

$$\hat{\mathbf{L}}_n(v) = \frac{1}{n} \sum_{i=1}^n f(\hat{s}, Z_i) \tag{31}$$

where $f(s, Z) = (s_1(Z) + h_1(Z) + s_2(Z) + h_2(Z) s_3(Z))^2$ and $\mathcal{D}_n := \{Z_i\}_{i=1}^n = \{X_i, Y_i\}_{i=1}^n \overset{iid}{\sim} p_{X,Y}$ is an i.i.d. dataset, and $\hat{s} = (\hat{s}_1, \hat{s}_2, \hat{s}_3)$ are the estimated scores using $\mathcal{D}_n$. We can expand the deviation between the estimated and true loss into two bounding terms

$$|\hat{\mathbf{L}}_n(v) - \mathbf{L}(v)| \le \left| \frac{1}{n} \sum_{i=1}^n f(\hat{s}, Z_i) - \mathbb{E}_Z f(\hat{s}, Z) \right| + |\mathbb{E}_Z f(\hat{s}, Z) - \mathbb{E} f(s, Z)| \tag{32}$$

$$\le \sup_{s \in \mathcal{B}(\mathcal{H}, M)} \left| \frac{1}{n} \sum_{i=1}^n f(s, Z_i) - \mathbb{E} f(s, Z) \right| + |\mathbb{E}_Z f(\hat{s}, Z) - \mathbb{E} f(s, Z)| \tag{33}$$

where for clarity $\mathcal{B}(\mathcal{H}, M) = \mathcal{B}(\mathcal{H}_1, M) \cup \mathcal{B}(\mathcal{H}_2, M) \cup \mathcal{B}(\mathcal{H}_3, M) \subset \mathcal{H}$, $f(s, z) = (s_1(z) + h_1(z) + s_2(z) + h_2(z) s_3(z))^2$, and $Z$ is an iid copy.

To analyze the LHS term, we first show that the function class $\{f(s, \cdot) : s \in \mathcal{B}(\mathcal{H}, M)\}$ is a *separable and complete Carathéodory family*. This requires that (i) $\mathcal{B}(\mathcal{H}, M)$ is a separable, complete metric space, and (ii) $s \mapsto f(s, z)$ is

continuous for every $z \in \mathcal{Z}$ (Steinwart, 2008). (i) Follows immediately from the properties of RKHS's and the fact that $\mathcal{B}(\mathcal{H}, M)$ is a closed ball in this space. (ii) Follows from the fact that $s_1, s_2, s_3, h_1, h_2$ are all bounded and continuous (note the boundedness of $s_1, s_2, s_3$ follows from the boundedness of the kernels $k_x, k_y$). This property also means that $\mathcal{F} := \{f(s, \cdot) : s \in \mathcal{B}(\mathcal{H}, M)\} \in L_\infty(\mathcal{Z})$ and $\sup_{s \in \mathcal{B}(\mathcal{H}, M)} \|f(s, \cdot)\|_\infty$. As a result, by Proposition 7.10 in Steinwart (2008) we have

$$\mathbb{E} \sup_{s \in \mathcal{B}(\mathcal{H}, M)} \left| \mathbb{E} f(s, Z) - \frac{1}{n} \sum_{i=1}^n f(s, Z_i) \right| \le 2 \mathbb{E} \mathrm{Rad}_{\mathcal{D}_n}(\mathcal{F}, n) \tag{34}$$

Where $\mathrm{Rad}_{\mathcal{D}_n}(\mathcal{F}, n) = \mathbb{E}_\sigma \sup_{s \in \mathcal{B}(\mathcal{H}, M)} \left| \frac{1}{n} \sum_{i=1}^n \sigma_i f(s, Z_i) \right|$ is the empirical Rademacher average (i.e., $\{\sigma_i\}_{i=1}^n \overset{iid}{\sim}$ Rad(1/2)). Now, note that we can expand $f$ as

$$f(s, Z) = (s_1(Z)^2 + s_2(Z)^2 + h_2(Z)^2 s_3(Z)^2 + 2 s_1(Z) s_2(Z) + 2 s_1(Z) s_3(Z) h_2(Z) + 2 s_2(Z) s_3(Z) h_2(Z) \\ + 2(s_1(Z) + s_2(Z) + s_3(Z) h_2(Z)) h_1(Z) + h_1(Z)^2) \tag{35}$$

If we substitute in this definition of $f$ into (34), we get the inequality.

$$\mathrm{Rad}_{\mathcal{D}_n}(\mathcal{F}, n) \le \mathrm{Rad}_{\mathcal{D}_n}(\mathcal{B}_{\mathcal{H}_1^2}, n) + \mathrm{Rad}_{\mathcal{D}_n}(\mathcal{B}_{\mathcal{H}_2^2}, n) + \mathrm{Rad}_{\mathcal{D}_n}(\{h_2\} \otimes \mathcal{B}_{\mathcal{H}_3^2}, n) + 2 \mathrm{Rad}_{\mathcal{D}_n}(\mathcal{B}_{\mathcal{H}_1} \otimes \mathcal{B}_{\mathcal{H}_2}, n) \\ + 2 \mathrm{Rad}_{\mathcal{D}_n}(\mathcal{B}_{\mathcal{H}_1} \otimes \mathcal{B}_{\mathcal{H}_3} \otimes \{h_2\}, n) + 2 \mathrm{Rad}_{\mathcal{D}_n}(\mathcal{B}_{\mathcal{H}_2} \otimes \mathcal{B}_{\mathcal{H}_3} \otimes \{h_2\}, n) + 2 \mathrm{Rad}_{\mathcal{D}_n}(\mathcal{B}_{\mathcal{H}_1} \otimes \{h_1\}, n) \\ + 2 \mathrm{Rad}_{\mathcal{D}_n}(\mathcal{B}_{\mathcal{H}_2} \otimes \{h_1\}, n) + 2 \mathrm{Rad}_{\mathcal{D}_n}(\mathcal{B}_{\mathcal{H}_3} \otimes \{h_1\} \otimes \{h_2\}, n) + \mathrm{Rad}_{\mathcal{D}_n}(\{h_1\} \otimes \{h_1\}, n) \tag{36}$$

Where in the above we use the shorthand $\mathcal{B}_\mathcal{H} = \mathcal{B}(\mathcal{H}, M)$. The tensor product spaces with singletons satisfy the property that if $\phi \in \mathcal{G} \otimes \{h\}$ where $\mathcal{G}$ is a space of functions and $\{h\}$ is a singleton, then there is some $g \in \mathcal{G}$ such that $\phi(z) = g(z) h(z), \forall z \in \mathcal{Z}$. Note that the above inequality follows from the fact that, for any two function classes $\mathcal{G}_1, \mathcal{G}_2$, we have by the triangle inequality,

$$\sup_{(g_1, g_2) \in \mathcal{G}_1 \times \mathcal{G}_2} \left| \frac{1}{n} \sum_{i=1}^n \sigma_i (g_1(Z_i) + g_2(Z_i)) \right| \le \sup_{g_1 \in \mathcal{G}_1} \left| \frac{1}{n} \sum_{i=1}^n \sigma_i g_1(Z_i) \right| + \sup_{g_2 \in \mathcal{G}_2} \left| \frac{1}{n} \sum_{i=1}^n \sigma_i g_2(Z_i) \right| \tag{37}$$

Now, note that every term in (36) is the Rademacher average of a closed ball in an RKHS of bounded functions. For the terms involving tensor-product spaces without singletons like $\{h_1\}$ this is by true by definition (since we work with tensor products of bounded RKHS's). To briefly show that this holds for the terms involving tensor product spaces with singletons (e.g. $\mathcal{H}_1 \otimes \{h_1\}$), note in general that if $\mathcal{H}_k$ is an RKHS with bounded kernel $k$, then so is $\mathcal{H}_k \otimes \{h_1\}$, if $h_1 \in C^0(\mathcal{Z}, \mathbb{R})$, because the evaluation functional $\delta_z : h \otimes h_1 \mapsto h(z) h_1(z)$ remains continuous: $\delta_z(h h_1) = \langle h, k(z, \cdot) \rangle_{\mathcal{H}_k} h_1(z) \le \|h_1\|_\infty k(z, z) \|h\|_{\mathcal{H}_k} \le \tilde{B} \|h\|_{\mathcal{H}_k}, \tilde{B} > 0$. Similarly, if $\|h\|_{\mathcal{H}_k} \le M$, then $\|h h_1\|_{\mathcal{H}_k} \le M$ also. This is because one can simply choose the norm of $\mathcal{H}_k \otimes \{h_1\}$ as the norm of $\mathcal{H}_k$, due to the boundedness of $h_1$.

Now, by Lemma 22 in Bartlett & Mendelson (2002), we have $\mathrm{Rad}_{\mathcal{D}_n}(\mathcal{B}(\mathcal{H}_k, M), n) < \frac{C}{\sqrt{n}}$ for some $C > 0$, where $\mathcal{B}(\mathcal{H}_k, M) = \{h \in \mathcal{H}_k : \|h\|_{\mathcal{H}_k} \le M\}$ and $\mathcal{H}_k$ is an RKHS with bounded kernel $k$. This means that

$$\mathbb{E} \sup_{s \in \mathcal{B}(\mathcal{H}, M)} \left| \mathbb{E} f(s, Z) - \frac{1}{n} \sum_{i=1}^n f(s, Z_i) \right| \le 2 \mathbb{E} \mathrm{Rad}_{\mathcal{D}_n}(\mathcal{F}, n) \le 2D / \sqrt{n} \tag{38}$$

for an appropriate constant $D > 0$, which gives us the desired result for the first term of Equation (33).

Now for the second term. For this section, all expectations are taken over $Z$ only (i.e. not $\hat{s}$). To start, we can expand out $f$.

$$\mathbb{E} f(s, Z) = \mathbb{E}[s_1(Z)^2 + s_2(Z)^2 + h_2(Z)^2 s_3(Z)^2 + 2 s_1(Z) s_2(Z) + 2 s_1(Z) s_3(Z) h_2(Z) + 2 s_2(Z) s_3(Z) h_2(Z) \\ + 2(s_1(Z) + s_2(Z) + s_3(Z) h_2(Z)) h_1(Z) + h_1(Z)^2] \tag{39}$$

Taking differences with the same quantity at $\hat{s}$ gives

$$\mathbb{E} f(s, Z) - \mathbb{E} f(\hat{s}, Z) = \mathbb{E}(s_1(Z)^2 - \hat{s}_1(Z)^2 + \mathbb{E}(s_2(Z)^2 - \hat{s}_2(Z)^2) + \mathbb{E} h_2(Z)^2 (s_3(Z)^2 - \hat{s}_3(Z)^2) \tag{40}$$
$$+ 2 \mathbb{E}[s_1(Z) s_2(Z) - \hat{s}_1(Z) \hat{s}_2(Z)] + 2 \mathbb{E}[(s_1(Z) s_3(Z) - \hat{s}_1(Z) \hat{s}_3(Z)) h_2(Z)] + 2 \mathbb{E}[(s_2(Z) s_3(Z) - \hat{s}_2(Z) \hat{s}_3(Z)) h_2(Z)] \\ + 2 \mathbb{E}[(s_1(Z) - \hat{s}_1(Z)) h_1(Z)] + 2 \mathbb{E}[(s_2(Z) - \hat{s}_2(Z)) h_1(Z)] + 2 \mathbb{E}[(s_3(Z) - \hat{s}_3(Z)) h_1(Z)] \tag{41}$$

Since $v \in C^1(\mathbb{R}^2, \mathbb{R})$, we know $h_1 < A_1, h_2 < A_2$ are bounded, where $A_1, A_2 > 0$ are constants. Using Jensen's inequality we therefore have

$$|\mathbb{E}f(s, Z) - \mathbb{E}f(\hat{s}, Z)| \leq \mathbb{E}|s_1(Z)^2 - \hat{s}_1(Z)^2| + \mathbb{E}|s_2(Z)^2 - \hat{s}_2(Z)^2| + A_1\mathbb{E}|s_3(Z)^2 - \hat{s}_3(Z)^2|$$
$$+ 2\mathbb{E}[s_1(Z)s_2(Z) - \hat{s}_1(Z)\hat{s}_2(Z)] + 2A_2\mathbb{E}[|s_1(Z)s_3(Z) - \hat{s}_1(Z)\hat{s}_3(Z)|] + 2A_2\mathbb{E}[(s_2(Z)s_3(Z) - \hat{s}_2(Z)\hat{s}_3(Z))]$$
$$+ 2A_1\mathbb{E}[(s_1(Z) - \hat{s}_1(Z)) + 2A_1\mathbb{E}[|s_2(Z) - \hat{s}_2(Z)] + 2A_1\mathbb{E}[(|s_3(Z) - \hat{s}_3(Z)|] \tag{42}$$

Which can be simplified using a change of notation to

$$|\mathbb{E}f(s, Z) - \mathbb{E}f(\hat{s}, Z)| \leq \|s_1^2 - \hat{s}_1^2\|_{L_1} + \|s_2^2 - \hat{s}_2^2\|_{L_1} + \|s_3^2 - \hat{s}_3^2\|_{L_1} \tag{43}$$
$$+ 2\|s_1 s_2 - \hat{s}_1\hat{s}_2\|_{L_1} + 2A_2\|s_1 s_3 - \hat{s}_1\hat{s}_3\|_{L_1} + 2A_2\|s_2 s_3 - \hat{s}_2\hat{s}_3\|_{L_1}$$
$$+ 2A_1\|s_1 - \hat{s}_1\|_{L_1} + 2A_1\|s_2 - \hat{s}_2\|_{L_1} + 2A_1\|s_3 - \hat{s}_2\|_{L_1} \tag{44}$$

where we define $s_i^2 : z \mapsto s_i(z)^2$ and $(s_i s_j) : z \mapsto s_i(z)s_j(z)$. There are two kinds of summands above: (i) $\|s_i - \hat{s}_i\|_{L_1}$ and (ii) $\|s_i s_j - \hat{s}_i\hat{s}_j\|_{L_1}$ for $i, j \in \{1, 2, 3\}$. All that remains is to bound each of these terms by sums of terms like $\|s_i - \hat{s}_i\|_{L_2}$. This is immediate for (i) by the properties of $L_p$ norms. To show (ii) we can simply use the triangle inequality and Cauchy Schwartz:

$$\|s_i s_j - \hat{s}_i\hat{s}_j\|_{L_1} = \|s_i(s_j - \hat{s}_j) + s_j(\hat{s}_i - s_i)\|_{L_1} \tag{45}$$
$$\leq \mathbb{E}|s_i(Z)||s_j(Z) - \hat{s}_j(Z)| + \mathbb{E}|s_j(Z)||(\hat{s}_i(Z) - s_i(Z))| \tag{46}$$
$$\leq \|s_i\|_{L_2}\|s_j - \hat{s}_j\|_{L_2} + \|s_j\|_{L_2}\|s_i - \hat{s}_i\|_{L_2} \tag{47}$$
$$\leq A_S(\|s_j - \hat{s}_j\|_{L_2} + \|s_i - \hat{s}_i\|_{L_2}) \tag{48}$$

Now, note that for any bounded positive-definite kernel $k : \mathcal{Z}^2 \to \mathbb{R}$ with associated RKHS $\mathcal{H}_k$, if $k < B$ we have $\|f\|_{L_2} \leq \|f\|_{\mathcal{H}_k}$, because $|f(z)|^2 = \langle k(z, \cdot), f\rangle_{\mathcal{H}_k}^2 \leq |k(z, z)|\|f\|_{\mathcal{H}_k}^2 \leq B$ . This means that

$$|\mathbb{E}f(s, Z) - \mathbb{E}f(\hat{s}, Z)| = \mathcal{O}\left(\sum_{i=1}^3 \|s_i - \hat{s}_i\|_{\mathcal{H}_i}\right) = \mathcal{O}_p(n^{-\frac{1}{\alpha}}). \tag{49}$$

Where the last equality follows from the convergence assumption of the score estimators in the theorem. Combining this result with the convergence result in Equation (38) for the first term of Equation (33) completes the proof.

$\square$

## C. Identifiability of Causal Direction with Bivariate Velocity Models

In order for the causal direction to be identifiable, it cannot be the case that $p(x, y)$ can be expressed in terms of a SCM from the same class in both causal directions. For the model in the $Y \to X$ direction, let $\tilde{v}(x, y)$ denote the velocity. Theorem 4.1 can be used to determine conditions under which the causal direction cannot be identified.

For convenience, let $\pi$ denote $\log p$, for example $\pi(x, y) = \log p(x, y)$, $\pi(y|x) = \log p(y|x)$, and so on. Starting with (14), taking a partial derivative in $y$ yields

$$\partial_y \partial_x \pi(x, y) = -\partial_y^2 v(y, x) - \partial_y v(y, x)\partial_y \pi(x, y) - v(y, x)\partial_y^2 \pi(x, y) . \tag{50}$$

Similarly, in other model direction, (14) becomes $\partial_y \pi(x, y) = -\partial_x \tilde{v}(x, y) - \tilde{v}(x, y)\partial_x \pi(x, y) + \partial_y \pi(y)$, so taking a derivative with respect to $x$, we have

$$\partial_x \partial_y \pi(x, y) = -\partial_x^2 \tilde{v}(x, y) - \partial_x \tilde{v}(x, y)\partial_x \pi(x, y) - \tilde{v}(x, y)\partial_x^2 \pi(x, y) . \tag{51}$$

Equating them, we find that the direction is not identifiable if and only if

$$\partial_y^2 v(y, x) + \partial_y v(y, x)\partial_y \pi(x, y) + v(y, x)\partial_y^2 \pi(x, y) = \partial_x^2 \tilde{v}(x, y) + \partial_x \tilde{v}(x, y)\partial_x \pi(x, y) + \tilde{v}(x, y)\partial_x^2 \pi(x, y) . \tag{52}$$

This proves Proposition 5.2.

To use this, we write the various partial derivatives of $\pi(x, y)$ in terms of the forward model. For a SCM with velocity $v$ and flow $\varphi$, so that the SCM is $y = \varphi_{x_0, x}(\epsilon_y)$, the log joint density can be written

$$\pi(x, y) = \pi(x) + \pi_0(\varphi_{x_0, x}^{-1}(y)) + \log|\partial_y \varphi_{x_0, x}^{-1}(y)| = \pi(x) + \pi_0(\varphi_{x, x_0}(y)) + \log|\partial_y \varphi_{x, x_0}(y)| \ ,$$

and therefore,

$$\partial_y \pi(x, y) = \dot{\pi}_0(\varphi_{x, x_0}(y)) \partial_y \varphi_{x, x_0}(y) + \partial_y \log|\partial_y \varphi_{x, x_0}(y)| \tag{53}$$

$$\partial_x \pi(x, y) = \dot{\pi}(x) + \dot{\pi}_0(\varphi_{x, x_0}(y)) \partial_x \varphi_{x, x_0}(y) + \partial_x \log|\partial_y \varphi_{x, x_0}(y)| \tag{54}$$

$$= \dot{\pi}(x) - \dot{\pi}_0(\varphi_{x, x_0}(y)) v(y, x) \partial_y \varphi_{x, x_0}(y) - \partial_y v(y, x) \tag{55}$$

$$\partial_y^2 \pi(x, y) = \ddot{\pi}_0(\varphi_{x, x_0}(y)) (\partial_y \varphi_{x, x_0}(y))^2 + \dot{\pi}_0(\varphi_{x, x_0}(y)) \partial_y^2 \varphi_{x, x_0}(y) + \partial_y^2 \log|\partial_y \varphi_{x, x_0}(y)| \tag{56}$$

$$\partial_x^2 \pi(x, y) = \ddot{\pi}(x) + \ddot{\pi}_0(\varphi_{x, x_0}(y)) (\partial_x \varphi_{x, x_0}(y))^2 + \dot{\pi}_0(\varphi_{x, x_0}(y)) \partial_x^2 \varphi_{x, x_0}(y) - \partial_x \partial_y v(y, x) \tag{57}$$

$$\partial_x \partial_y \pi(x, y) = \ddot{\pi}_0(\varphi_{x, x_0}(y)) (\partial_y \varphi_{x, x_0}(y))(\partial_x \varphi_{x, x_0}(y)) + \dot{\pi}_0(\varphi_{x, x_0}(y))(\partial_x \partial_y \varphi_{x, x_0}(y)) + \partial_y \partial_x \log|\partial_y \varphi_{x, x_0}(y)| \tag{58}$$

$$= -\ddot{\pi}_0(\varphi_{x, x_0}(y)) v(y, x)(\partial_y \varphi_{x, x_0}(y))^2 - \dot{\pi}_0(\varphi_{x, x_0}(y)) \partial_y (v(y, x) \partial_y \varphi_{x, x_0}(y)) - \partial_y^2 v(y, x) \tag{59}$$

We have used the identities $\partial_x \varphi_{x, x_0}(y) = -v(y, x) \partial_y \varphi_{x, x_0}(y)$ and $\partial_x \log|\partial_y \varphi_{x, x_0}(y)| = -\partial_y v(y, x)$ to simplify some of the expressions. Substituting these into (52), if we view $\pi_0$, $v$, $\varphi$, and $\pi(x)$ as given (specified by nature), the result is a PDE for the reverse model velocity, $\tilde{v}$. Alternatively, as is common in the literature (Peters & Bühlmann, 2014), if we allow $\pi(x)$ to vary then we might manipulate some combination of Equations (14) and (50) to (52) to obtain a differential equation for $\pi(x)$ in terms of only the fixed forward model.

A more thorough general analysis of identifiability is beyond the scope of this work, though we analyze below the special cases of ANMs and LSNMs. In doing so, we obtain a new characterization of non-identifiability that holds uniformly over the model class. However, even in that somewhat simple extension of ANMs, the characterizing equation is much more complicated and does not yield an easy interpretation.

### C.1. Additive noise models

For ANMs, write the models as $Y = m(X) + \epsilon_y$ with $\epsilon_y \sim p_0$, and $X = \tilde{m}(Y) + \epsilon_x$ with $\epsilon_x \sim \tilde{p}_0$. As shown in the main text, $v(y, x) = \dot{m}(x)$. Putting this into (52) yields

$$\dot{m}(x) \partial_y^2 \pi(x, y) = \dot{\tilde{m}}(y) \partial_x^2 \pi(x, y) \ .$$

Hence, the structural functions are mutually constrained, with $\dot{\tilde{m}}(y)$ satisfying

$$\dot{\tilde{m}}(y) = \dot{m}(x) \frac{\partial_y^2 \pi(x, y)}{\partial_x^2 \pi(x, y)} \ . \tag{60}$$

We observe that in the special case of linear ANMs, $\dot{m}(x) = a$ and $\dot{\tilde{m}}(y) = b$, so that (60) implies that each of $\partial_y^2 \pi(x, y) = \partial_y^2 \pi(y|x)$ and $\partial_x^2 \pi(x, y) = \partial_x^2 \pi(x|y)$ must be constant and therefore Gaussian, i.e., $b/a = \sigma_x^2 / \sigma_y^2$. We also see that if $\pi(x, y)$ is jointly Gaussian then the ANM is not identifiable if and only if $\dot{m}(x)/\dot{\tilde{m}}(y)$ is a constant, i.e., each of the model directions is linear.

Continuing from (60) by taking another partial derivative in $x$, we find that

$$\partial_x \frac{\dot{m}(x) \partial_y^2 \pi(x, y)}{\partial_x^2 \pi(x, y)} = 0 \ .$$

Moreover,

$$\pi(x, y) = \pi(x) + \pi_0(y - m(x)) \ ,$$

so that

$$\partial_y \pi(x, y) = \dot{\pi}_0(y - m(x))$$

$$\partial_y^2 \pi(x, y) = \ddot{\pi}_0(y - m(x))$$

$$\partial_x \pi(x, y) = \dot{\pi}(x) - \dot{\pi}_0(y - m(x)) \dot{m}(x)$$

$$\partial_x^2 \pi(x, y) = \ddot{\pi}(x) + \ddot{\pi}_0(y - m(x))(\dot{m}(x))^2 - \dot{\pi}_0(y - m(x)) \ddot{m}(x)$$

Carrying out the algebra, we find

$$\dddot{\pi}(x) = \ddot{\pi}(x)G(x,y) + H(x,y) \,, \tag{61}$$

where

$$G(x,y) = \frac{\dddot{m}(x)}{\dot{m}(x)} - \frac{\dot{m}(x)\,\dddot{\pi}_0(y - m(x))}{\ddot{\pi}_0(y - m(x))}$$

$$H(x,y) = -2\ddot{\pi}_0(y - m(x))\ddot{m}(x)\dot{m}(x) + \dot{\pi}_0(y - m(x))\dddot{m}(x)$$

$$+ \frac{\dot{\pi}_0(y - m(x))\,\dddot{\pi}_0(y - m(x))\ddot{m}(x)\dot{m}(x)}{\ddot{\pi}_0(y - m(x))} - \frac{\dot{\pi}_0(y - m(x))(\ddot{m}(x))^2}{\dot{m}(x)} \,.$$

This is the same differential equation obtained by Hoyer et al. (2008) in their analysis of identifiability using ANM models.

Alternatively, Eq. (6) from (Hoyer et al., 2008), which also leads to the identifying differential equation (61), can be obtained directly from the continuity equation (14) for the backward model $Y \to X$. In that case, (51) yields

$$\partial_x \partial_y \pi(x,y) = -\dot{\tilde{m}}(y)\partial_x^2 \pi(x,y).$$

Hence, solving for $1/\dot{\tilde{m}}(y)$ and differentiating with respect to $x$,

$$\partial_x \left( \frac{\partial_x \partial_y \pi(x,y)}{\partial_x^2 \pi(x,y)} \right) = 0 \,,$$

which is Eq. (6) in (Hoyer et al., 2008), from which the rest of their identifiability results on ANMs follow.

### C.2. Location-Scale Noise Models

For LSNMs with $y = m(x) + e^{h(x)}\epsilon_y$ and $x = \tilde{m}(y) + e^{\tilde{h}(y)}\epsilon_x$, we have that $v(y,x) = \dot{m}(x) + \dot{h}(x)(y - m(x))$, and similarly $\tilde{v}(x,y) = \dot{\tilde{m}}(y) + \dot{\tilde{h}}(y)(x - \tilde{m}(y))$. Therefore, (51) yields

$$-\partial_x \partial_y \pi(x,y) = \dot{\tilde{h}}(y)\partial_x \pi(x,y) + (\dot{\tilde{m}}(y) + \dot{\tilde{h}}(y)(x - \tilde{m}(y)))\partial_x^2 \pi(x,y) \,.$$

Dividing by $\partial_x^2 \pi(x,y)$ and differentiating with respect to $x$,

$$-\partial_x \left( \frac{\partial_x \partial_y \pi(x,y)}{\partial_x^2 \pi(x,y)} \right) = \dot{\tilde{h}}(y)\left( 1 + \partial_x \left( \frac{\partial_x \pi(x,y)}{\partial_x^2 \pi(x,y)} \right) \right) \,.$$

Therefore,

$$\partial_x \left[ \partial_x \left( \frac{\partial_x \partial_y \pi(x,y)}{\partial_x^2 \pi(x,y)} \right) \bigg/ \left( 1 + \partial_x \left( \frac{\partial_x \pi(x,y)}{\partial_x^2 \pi(x,y)} \right) \right) \right] = 0 \,.$$

Carrying out the differentiation and simplifying yields (suppressing the $(x,y)$ arguments of $\pi$),

$$[3(\partial_x^3 \pi)^2 - 2(\partial_x^2 \pi)(\partial_x^4 \pi)](\partial_x \partial_y \pi) + [(\partial_x \pi)(\partial_x^4 \pi) - 3(\partial_x^2 \pi)(\partial_x^3 \pi)](\partial_x^2 \partial_y \pi) + [2(\partial_x^2 \pi) - (\partial_x \pi)(\partial_x^3 \pi)](\partial_x^3 \partial_y \pi) = 0 \,.$$

Since by using the forward model we have

$$\pi(x,y) = \pi(x) + \pi_0\left( e^{-h(x)}(y - m(x)) \right) - \dot{h}(x) \,,$$

in principle this can be solved to characterize, for fixed $\pi_0, h, m$ the set of $\pi(x)$ for which the causal direction is not identifiable.

Alternatively, substituting

$$\partial_x \partial_y \pi(x,y) = \dot{h}(x)\partial_y \pi(x,y) + (\dot{m}(x) + \dot{h}(x)(y - m(x)))\partial_y^2 \pi(x,y)$$

yields a different but equivalent equation.

Table 4. Parameter settings for synthetic benchmarks.

| Benchmark | $\theta$ | $f_\theta(x, \epsilon_y)$ | $\sigma_\theta$ | $\sigma_y$ |
|---|---|---|---|---|
| Velocity | $\theta \in \mathbb{R}^6$ | $\epsilon_y + \int\limits_{x_0}^{x} \theta^\top \Phi(y(u), u) du,$ | 1 | 1 |
| | | $\Phi(y, u)^\top = \begin{bmatrix} 1 \\ \sin(u) \\ \sin(y) \\ \cos(u) \\ \cos(y) \\ \sin(x+y) \end{bmatrix}$ | | |
| Sigmoid | $\theta_a, \theta_b, \theta_c, \theta_d$: 2x64 MLP weights | $c(x) + e^{-d(x)^2} \Phi^{-1}(\text{sigmoid}(a(x) + e^{-b(x)^2} \epsilon_y))$ | 0.2 | 3 |
| ANM | $\theta_m$ 3x64 MLP weights | $m(x) + \epsilon_y$ | 0.2 | 0.2 |
| LSNM | $\theta_m, \theta_h$ 2x64 MLP weights | $m(x) + (e^{-h(x)^2} + 0.2)\epsilon_y$ | 0.2 | 0.2 |

## D. Experiment and Simulation Details

We describe extra details of the experiment and simulation here. All experiments are conducted using the JAX library (Bradbury et al., 2018).

### D.1. Synthetic Data Generation

All synthetic benchmarks can be described as generating from an SCM $Y = f_\theta(X, \epsilon_y)$. In each case 100 samples are drawn from $\theta \sim \mathcal{N}(0, \sigma_\theta^2)$ to generate the 100 datasets. The noise distributions $X, \epsilon_y$ are generated as follows.

$$\xi_x, \xi_y \sim \mathcal{N}(0, I_2), \tag{62}$$

$$X = T_{\theta_t}(\xi_x), \epsilon_y = \sigma_y T_{\theta_t}, (\xi_y) \tag{63}$$

where $\sigma_y$ is a noise scaling parameter that represents the signal-to-noise ratio (larger $\sigma_y$ indicates noisier data). See Table 4 for specifics in each benchmark. $T_{\theta_t}$ are randomly sampled (for each dataset in the benchmark) triangular monotonic increasing (TMI) maps parametrized as follows:

$$T_{\theta_t}(x) = \int_0^x \text{softplus}(f_{\theta_t}(x)) dx, \tag{64}$$

and $\theta_t \sim \mathcal{N}(0, 0.3^2)$ are 3x64 MLP parameters.

**Analytic Score Calculation** For Section 7.3, we additionally designed ANM and LSNM datasets with Gaussian noise variables, $X \sim \mathcal{N}(0, 1), \epsilon_y \sim \mathcal{N}(0, \sigma_y^2)$. Other settings are as in Table 4. The Gaussian noise variables allow for an analytic score calculation of the marginal and joint score functions, as follows:

$$s_x(x) = s_{\text{gaussian}}(x) = -x. \tag{65}$$

Denote the mechanism by $f_x(\epsilon_y)$. We have

$$s_x(x, y) = \partial_x \log p(x, y) = \partial_x \log p(x) + \partial_x \log p(y \mid x) \tag{66}$$

$$= -x + \partial_x \log p_{\epsilon_y}(f_x^{-1}(y)) + \partial_x (\log \partial_y f_x^{-1}(y)) \tag{67}$$

$$= -x + f_x^{-1}(y)\partial_x f_x^{-1}(y) + \frac{\partial_{xy} f_x^{-1}(y)}{\partial_y f_x^{-1}(y)}, \tag{68}$$

and

$$s_y(x, y) = \partial_y \log p(x, y) = \partial_y \log p(x) + \partial_y \log p(y \mid x) \tag{69}$$

$$= \partial_y \log p_{\epsilon_y}(f_x^{-1}(y)) + \partial_y (\log \partial_y f_x^{-1}(y)) \tag{70}$$

$$= f_x^{-1}(y)\partial_y f_x^{-1}(y) + \frac{\partial_{yy} f_x^{-1}(y)}{\partial_y f_x^{-1}(y)}, \tag{71}$$

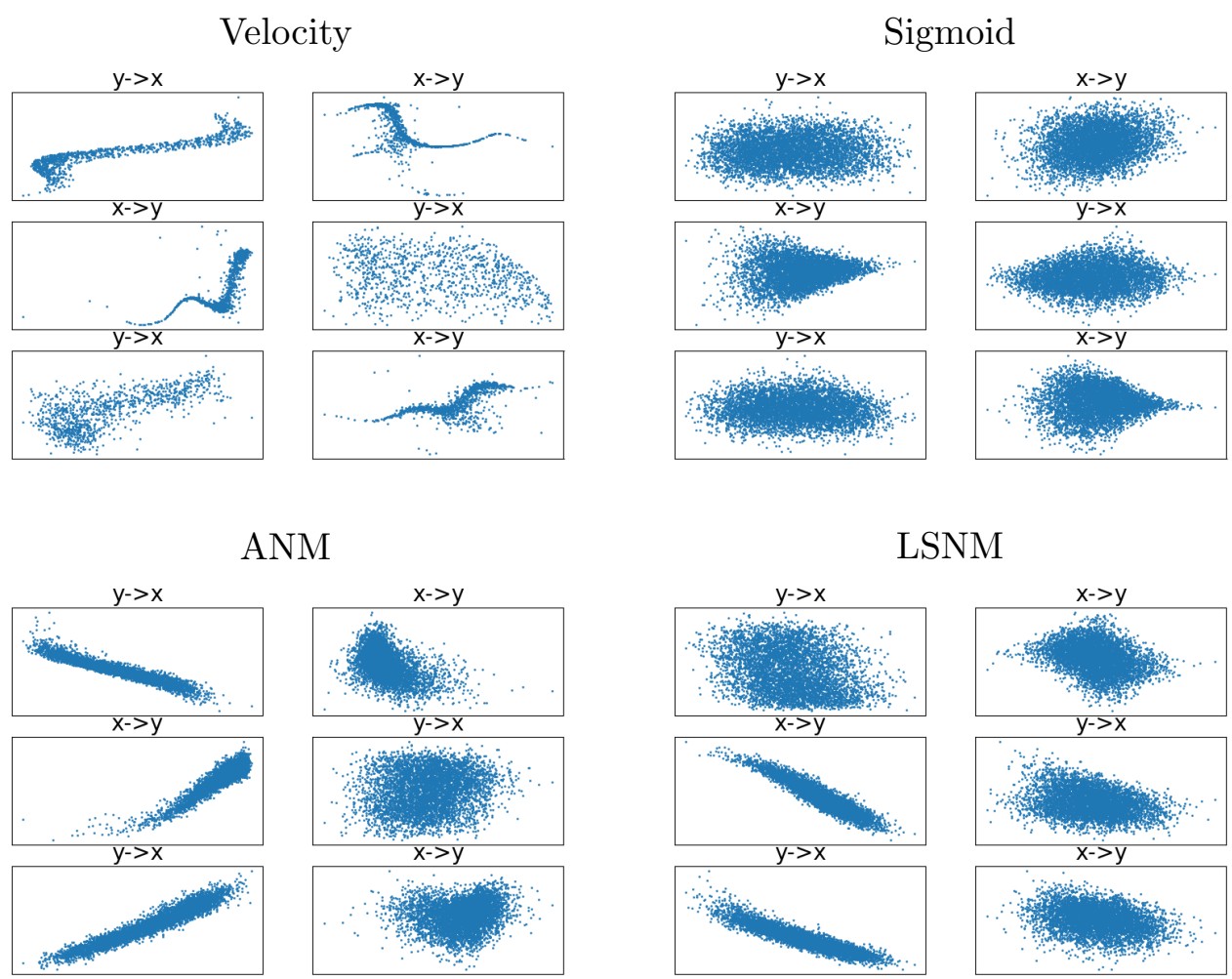

*Figure 6.* Example datasets from each synthetic benchmark.

where the derivatives are calculated with automatic differentiation. Finally, the marginal density $y$ is given as

$$p(y) = \int_x p(y \mid x)p(x)dx, \tag{72}$$

and the score is approximated by Monte carlo approximation of the integral and automatic differentiation of the log-density.

### D.2. Velocity Parametrization and Training

We distinguish two types of models: those where the velocity function is a linear combination of basis functions, and those where the velocity function are neural network based.

For the basis models, we use the Adam optimizer with a base learning rate of $0.1$, scaled by a factor of $1/\log(\#\text{ of parameters})$.

The basis functions we use are as follows:

$$\Phi_{\text{lin}}(y, x) = \begin{bmatrix} 1 & x & y \end{bmatrix}, \quad \Phi_{\text{quad}}(y, x) = \begin{bmatrix} 1 & x & y & x^2 & y^2 & xy \end{bmatrix}. \tag{73}$$

For each of these, our experiments also include appending the following exponential terms:

$$\Phi_{\text{exp}}(y, x) = \begin{bmatrix} e^{-x^2} & e^{-y^2} & e^{-(x^2+y^2)} \end{bmatrix}. \tag{74}$$

The linear basis and quadratic basis models are hence parametrized by $K = 3, 6$ real-valued parameters, respectively. When the exponential terms are added, the parameter count is increased further by $3$.

All neural networks involved are 3 layer fully connected MLPs with a hidden size of $64$, and $\tanh$ activation functions. For the ANM, we directly parametrize the velocity (i.e., $\dot{m}$) as an MLP, while for the LSNM, we parametrize the functions $m$ and $h$, and evaluate their derivatives using automatic differentiation to obtain the LSNM velocity (see Table 1 for the specific form).

### D.3. Score Estimation

For the Stein score estimate, we use the suggested regularization parameter $\lambda = 0.1$ (Li & Turner, 2018). For the KDE, we use a regularization parameter of $\epsilon = n^{-2}$ as suggested in (Wibisono et al., 2024). For the KDE estimator, we found that the standard Silverman rule of thumb (Silverman, 2018) for obtaining the bandwidth works well, as the optimal bandwidths as proposed in (Wibisono et al., 2024) include unknown constants.

### D.4. Benchmarks

For the benchmarks in Table 3, we also remove the data points corresponding to the most extreme 5% of marginal values as an automated procedure, following score estimation. We found that this improved causal discovery performance especially for the real data in the Tübingen dataset. Furthermore, all other benchmarks have a sample size of $n = 1000$ besides the Tübingen dataset. To minimize hyperparameter search, all datasets are sub-sampled, or re-sampled if necessary, to a uniform size of $n = 1000$.

**Continuous Tübingen** In addition to the datasets with binary variables (pairs 47, 70, 107) and multivariate settings ("pairs" 52, 53, 54, 55, 71, and 105), which are filtered by default, we also identified 29 additional datasets with discrete variables that we expected score estimation to fail on, for example, the cause variable in pair 5 is integer-valued (# of tree rings). The full set of removed pairs are as follows: (5,6,7,8,9,10,11,13,14,15,16,26,27,28,29,32,33,34,35,36,37,47,70,85,94,95,99,105,107).

## E. Additional Figures and Results

### E.1. Synthetic Results

Here we present additional results on our synthetic dataset: Table 5 is the KDE counterpart to Table 2 in the main text, and Table 6 contains results when subsampled to $n = 1000$.

*Table 5.* KDE results on synthetic data ($n = 5000$).

| **Model** | Velocity | Sigmoid | ANM | LSNM |
|---|---|---|---|---|
| B-LIN | 68 (59) | 30 (27) | 45 (29) | 40 (32) |
| B-QUAD | 65 (61) | 33 (29) | 30 (19) | 15 (5) |
| V-ANM | 46 (33) | 25 (15) | 31(18) | 39 (26) |
| V-LSNM | 65 (78) | 47 (54) | 37 (33) | 38 (38)) |
| V-NN | 69 (65) | 56 (48) | 30 (24) | 36 (18) |

*Table 6.* Results on synthetic data (subsampled to $n = 1000$).

| **Model** | Velocity | | Sigmoid | | ANM | | LSNM | |
|---|---|---|---|---|---|---|---|---|
| | KDE | STEIN | KDE | STEIN | KDE | STEIN | KDE | STEIN |
| B-LIN | 68 (59) | 87 (96) | 30 (27) | 72 (88) | 35 (22) | 39 (30) | 40 (32) | 49 (60) |
| B-QUAD | 65 (57) | 88 (97) | 25 (28) | 67 (84) | 30 (19) | 49 (53) | 15 (5) | 51 (58) |
| V-ANM | 46 (33) | 81 (96) | 25 (15) | 40 (23) | 31 (18) | 77 (85) | 39 (26) | 52 (54) |
| V-LSNM | 65 (78) | 87 (96) | 47 (54) | 69 (86) | 37 (32) | 73 (82) | 38 (38) | 66 (64) |
| V-NN | 69 (65) | 90 (98) | 56 (49) | 58 (69) | 30 (24) | 58 (66) | 36 (18) | 48 (57) |
| LOCI (HSIC) | 40 (30) | | 70 (83) | | 91 (98) | | 69 (86) | |
| LOCI (Lik) | 46 (66) | | 49 (61) | | 44 (52) | | 31 (31) | |
| CDS | 46 (42) | | 28 (13) | | 90 (98) | | 48 (48) | |
| IGCI | 66 (76) | | 53 (67) | | 34 (22) | | 32 (20) | |
| RECI | 36 (26) | | 18 (8) | | 36 (34) | | 43 (56) | |
| CGCI | 48 (42) | | 66 (71) | | 71 (88) | | 55 (69) | |

## E.2. Additional Benchmarks

In Table 7, we present results on the benchmarks of Tagasovska et al. (2020). These are variants of ANM/LSNMs with Gaussian noise variables; the methods that assume Gaussian noise perform well in these settings.

*Table 7.* Results on the benchmark data of Tagasovska et al. (2020).

| **Model** | AN | | AN-s | | LS | | LS-s | | MNU | |
|---|---|---|---|---|---|---|---|---|---|---|
| | KDE | STEIN | KDE | STEIN | KDE | STEIN | KDE | STEIN | KDE | STEIN |
| B-LIN | 76 (87) | 14 (5) | 97 (100) | 17 (5) | 63 (79) | 19 (25) | 84 (96) | 52 (53) | 68 (79) | 52 (51) |
| B-QUAD | 98 (100) | 21 (8) | 98 (100) | 16 (4) | 89 (99) | 22 (25) | 92 (99) | 48 (53) | 96 (100) | 57 (71) |
| V-ANM | 93 (99) | 68 (85) | 73 (91) | 59 (67) | 77 (93) | 63 (71) | 57 (75) | 53 (72) | 55 (59) | 10 (2) |
| V-LSNM | 93 (98) | 44 (40) | 89 (98) | 31 (19) | 90 (99) | 69 (86) | 81 (92) | 62 (82) | 61 (68) | 41 (40) |
| V-NN | 97 (100) | 22 (6) | 94 (100) | 33 (17) | 84 (97) | 19 (20) | 74 (92) | 63 (79) | 55 (53) | 53 (64) |
| LOCI (HSIC) | 100 (100) | | 100 (100) | | 95 (99) | | 89 (97) | | 100(100) | |
| LOCI (Lik) | 100 (100) | | 100 (100) | | 100 (100) | | 100 (100) | | 100 (100) | |
| CDS | 99 (100) | | 92 (99) | | 77 (87) | | 6 (0) | | 66 (70) | |
| IGCI | 91 (98) | | 95 (100) | | 90 (98) | | 92 (98) | | 82 (93) | |
| RECI | 19 (6) | | 32 (18) | | 28 (14) | | 43 (43) | | 23 (7) | |
| CGCI | 100 (100) | | 95 (99) | | 100 (100) | | 85 (94) | | 95 (99) | |

## E.3. Sample Size Experiments with Known Score

Here, we report full tables of results corresponding to Figure 4 and Figure 5 in the main text, which is Table 8. We also repeated the same experiment for a well-specified LSNM, which is reported in Table 9. Finally, Figure 7 shows the visual effects of increasing the sample size, see Figure 3 for its counterpart when using the ground truth score. Note score estimation here refers to using the Stein score estimator with the Gaussian kernel.

*Table 8.* Known score sample size experiment results for ANM data. For MSE/GoF, values shown following the convention computed over the 100 datasets in the benchmark: Median (Q1, Q3).

| $n$ | Success (AUDRC) | MSE (Cause) | MSE (Effect) | MSE (Joint) | GoF (Causal) | GoF (Anticausal) |
|---|---|---|---|---|---|---|
| 100 | 55 (58) | 0.26 (0.17, 0.36) | 0.36 (0.18, 0.39) | 2.66 (1.46, 4.49) | 0.39 (0.35, 0.46) | 0.42 (0.36, 0.51) |
| 500 | 61 (75) | 0.08 (0.06, 0.12) | 0.09 (0.06, 0.14) | 2.27 (1.05, 4.10) | 0.29 (0.24, 0.32) | 0.31 (0.27, 0.36) |
| 1000 | 74 (88) | 0.06 (0.04, 0.08) | 0.05 (0.04, 0.08) | 2.17 (0.99, 4.26) | 0.23 (0.19, 0.27) | 0.26 (0.23, 0.30) |
| 2500 | 83 (96) | 0.03 (0.01, 0.04) | 0.03 (0.02, 0.04) | 2.07 (0.94, 4.20) | 0.18 (0.15, 0.20) | 0.22 (0.19, 0.28) |
| 5000 | 86 (97) | 0.02 (0.01, 0.03) | 0.02 (0.01, 0.03) | 2.06 (0.92, 4.13) | 0.14 (0.12, 0.16) | 0.19 (0.16, 0.23) |
| 10000 | 91 (96) | 0.02 (0.01, 0.04) | 0.02 (0.01, 0.05) | 2.03 (0.90, 4.21) | 0.12 (0.11, 0.14) | 0.19 (0.16, 0.23) |

*Table 9.* Sample size experiment results for LSNM data. For MSE/GoF, values shown following the convention computed over the 100 datasets in the benchmark: Median (Q1, Q3).

| $n$ | Success (AUDRC) | MSE (Cause) | MSE (Effect) | MSE (Joint) | GoF (Causal) | GoF (Anticausal) |
|---|---|---|---|---|---|---|
| 100 | 59 (51) | 0.25 (0.16, 0.43) | 0.29 (0.16, 0.43) | 1.28 (0.85, 1.92) | 0.65 (0.46, 0.87) | 0.67 (0.50, 0.83) |
| 500 | 53 (65) | 0.09 (0.05, 0.13) | 0.09 (0.06, 0.14) | 0.85 (0.65, 1.17) | 0.31 (0.27, 0.37) | 0.33 (0.28, 0.40) |
| 1000 | 70 (78) | 0.05 (0.03, 0.07) | 0.05 (0.03, 0.08) | 0.72 (0.59, 1.13) | 0.25 (0.22, 0.29) | 0.27 (0.24, 0.31) |
| 2500 | 67 (81) | 0.03 (0.02, 0.03) | 0.03 (0.02, 0.04) | 0.64 (0.49, 1.05) | 0.20 (0.17, 0.22) | 0.21 (0.18, 0.25) |
| 5000 | 73 (87) | 0.02 (0.01, 0.02) | 0.02 (0.01, 0.03) | 0.61 (0.47, 1.08) | 0.16 (0.14, 0.18) | 0.18 (0.16, 0.21) |
| 10000 | 80 (91) | 0.02 (0.01, 0.03) | 0.02 (0.01, 0.04) | 0.58 (0.45, 1.01) | 0.14 (0.13, 0.17) | 0.17 (0.14, 0.21) |

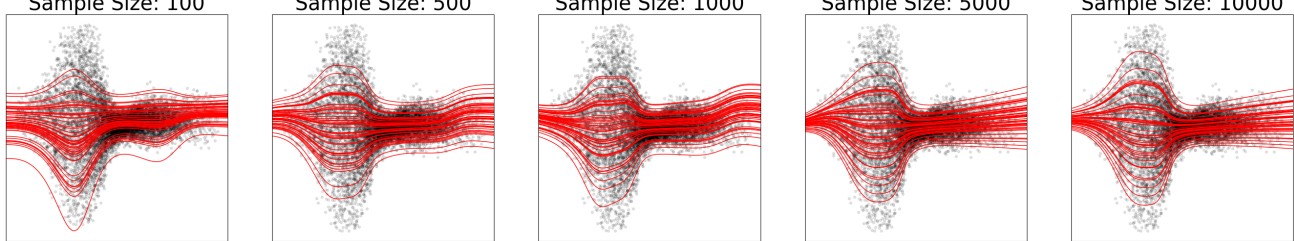

*Figure 7.* The estimated causal curves better resemble the ground truth as the score estimation improves in a well-specified LSNM.

