# OpenReview forum: "Distinguishing Cause from Effect with Causal Velocity Models"
_ICML.cc/2025/Conference — ICML 2025 poster_

### Official Review · Reviewer_H4Vs · 2025-03-12

**Overall Recommendation:** 4

**Summary:**

The authors proposed a novel solution to the bivariate causal discovery problem. The key idea is to view the SCM as a flow. The flow model is learned by posing the continuity constraints (minimizing an objective that forces the continuity equation). The value of this objective is further used to decide the causal direction (a smaller violation of the continuity suggests the causal direction).

**Claims And Evidence:**

Experimental evidence is not sufficient to support the claims. For example, I'd like to see if the optimized loss (in the causal direction) will be close to zero when the sample size goes to infinity.

**Essential References Not Discussed:**

I believe the following paper should be discussed.

Tu, Ruibo, et al. "Optimal transport for causal discovery." International Conference on Learning Representations, ICLR, 2022.

Additionally, there is an extensive body of research on bivariate causal discovery, including methods such as Conditional Divergence-based Causal Inference (CDCI) and Maximal Correlation-based PNL. The author may wish to engage in a comparative analysis with these approaches.

**Experimental Designs Or Analyses:**

I have several suggestions for improving the experiment presentation.

1. I don't think LOCI is the only one "SOTA" method. The comparison with other methods is necessary rather than referring to other papers. You may put the results in the supplementary.

2. On synthetic data. I don't think the authors should fix the noise level. Some methods may be sensitive to the noise level, say NOTEARS.

3. I'd like to see how the sample size will affect the accuracy of your methods of causal discovery.

4. Will the optimization of the loss be an issue? I'm not sure how fast the algorithm is.

**Methods And Evaluation Criteria:**

The methods and evaluation criteria are acceptable.

**Other Comments Or Suggestions:**

What makes me confused:
1. Notations in sec.2.3 are not clearly explained. $s$ for start time? $t$ for termination? I often got confused with $x$ and $t$ in this paper.

2. "Scores" in Section 4. What do the "scores" refer to? It seems the "scores" are term from the "score function" in Statistics rather than the causal score for distinguish the cause and effect. I suggest that the authors point out the causal score and its physical meaning explicitly in this section. To me, the physical meaning is simply the violation magnitudes of the continuity equation.

3. Notations like $\dot{m}$ are not defined. In convention, it refers to the first-order derivative w.r.t time. Is that true here?

In Figure 2, please consider showing the axis names and the SCM expression for each subfigure. BTW, I'm not sure what the purpose of showing quadratic and quadratic LSNM is.

**Other Strengths And Weaknesses:**

Strengths:
- The flow view of SCM is comprehensive.



Weaknesses:
-  The method relies on the accuracy of existing density estimation methods.

**Questions For Authors:**

1. Take ANM for example, can we regard the flow model as another representation? If so, the difference of the proposed method lies in how to learn the underlying functions and the decision criterion (proposed loss / HSIC test). Is it possible to furthur analysis the advantage of the proposed method compared to ANM? More effective learning or the decision rule is better?
2. Experiments when increasing/decreasing sample size.
3. Comparison with independent test-based methods and other recent methods.
4. Is the objective easy to optimize? What about the learning time? Hope to see some discussion.


I've updated my score accordingly.

**Relation To Broader Scientific Literature:**

TBD

**Theoretical Claims:**

I only read through the claims but didn't check the proofs.

---

> ### Author Rebuttal · Authors · 2025-03-31
>
> We thank the reviewer for their critical reading of our paper. The reviewer raises a few concerns and points of confusion that we believe can be addressed.
>
> **Reliance on score/density estimate**: Please see [response to reviewer 9rbC](https://openreview.net/forum?id=gV01DWTFTc&noteId=yMKZKiiUzH).
>
> **Questions for Authors**:
> 1. **Do flows represent ANMs?** Yes. We can regard the velocity/flow parameterization as a new way to express SCMs. ANMs, PNLs, and LSNMs are special cases (see Table 1). __The results in Tables 2-3 refer to the velocity parametrization/estimation of ANM/LSNM__. We can also express new SCMs easily via their velocities (see Table 1, Fig. 2, and Sec. D.2 for examples). In fact, Thm. 3.2 tells us that bijective SCMs and velocity models are in one-to-one correspondence. One of the advantages of our method is that entirely new classes of SCMs based on velocities can be fit and used for causal inference/discovery. ANMs are known or suspected to be mis-specified in many settings, in which case they may not be reliable for causal inference/discovery.
> 1. **Performance of method with varying sample sizes**: The method is sensitive to sample size mainly through the accuracy of score estimation. Given the exact score, Eq. 14 holds pointwise, so there is no statistical element remaining in minimizing Eq. 18. We designed an experiment to evaluate this empirically. Please see our [response to reviewer 9rbC, Tables 9rbC.1 and 9rbC.2](https://openreview.net/forum?id=gV01DWTFTc&noteId=yMKZKiiUzH) for details.
> 1. **Comparisons to additional methods**: We have evaluated ANM + independence tests and other methods from the `cdt` package, as well as CGCI, on the Tuebingen dataset (after filtering out discrete/ordinal datatsets, for which our model is mis-specified; see our response to reviewer sNSP) as well as the three simulated datasets in the paper (Table H4Vs.1 below). We find that our method performs best overall, although IGCI works well due to the Gaussian noise when generating the data.
> 1. **Difficulty of optimizing loss**: Please see our [response to reviewer xsfW](https://openreview.net/forum?id=gV01DWTFTc&noteId=pI8MqzAGXf) about computational considerations and optimization.
>
> **Table H4Vs.1**: Performance of our method compared to other methods
> | Method | Tuebingen (continuous distns. only) | Periodic | Sigmoid | Velocity |
> |---|---|---|---|---|
> | B-QUAD + KDE | 83% (0.89) | 99% (1.0) | 72% (0.87) | 95% (0.99) |
> | B-LIN + KDE | 79% (0.89)  | 98% (1.0) | 71% (0.82) | 95% (1.0) |
> | LOCI (best setting) | 63% (0.66)  | 86% (0.95) | 50% (0.72) | 61% (0.87) |
> | ANM | 60% (0.59) | 17% (0.10)  | 37% (0.25) | 12% (0.02) |
> | CDS | 61% (0.56) | 22% (0.09)  | 20% (0.08) | 12% (0.02) |
> | IGCI (Gaussian) | 53% (0.65) | 96% (0.99)  | 65% (0.81) | 87% (0.97) |
> | IGCI (Uniform) | 63% (0.67) | 0% (0)  | 16% (0.08) | 94% (0.99) |
> | RECI | 71% (0.88) | 0% (0) | 11% (0.03) | 92% (0.99) |
> | CGCI (best setting) | 61% (0.57) | 47% (0.37) | 65% (0.66) | 72% (0.74) |
>
> **Questions on Experimental Design/Analysis**
> - **LOCI not only SOTA method.** See answer 3 above.
> - **Noise level:**  To our knowledge, methods such as NOTEARS have been shown to be sensitive to marginal variances, in particular where the variance in the effect is larger than that of the cause. Following recent convention in the field first established by [Reisach et al., 2021], we standardize the data prior to causal discovery. Note that we use a noise scale of 3 when generating data to reduce the signal-to-noise ratio and increase the difficulty of causal discovery, but this affects all methods equally after standardization.
> - **Sample size.** See answer 2 above.
> - **Optimization of loss.** See answer 4 above.
>
>
> **Questions on Claims and Evidence**: **Optimized loss in causal direction as sample size diverges.** Please see [Table 9rbC.1](https://openreview.net/forum?id=gV01DWTFTc&noteId=yMKZKiiUzH), which shows that when the exact score is known and used, the optimized loss is small even for relatively small $n$, and appears to tend towards 0 for the causal direction, while it stays an order of magnitude larger in the anti-causal direction. For estimated scores, [Table 9rbC.2](https://openreview.net/forum?id=gV01DWTFTc&noteId=yMKZKiiUzH) shows a similar trend towards increasing discovery accuracy as score estimation improves with increasing $n$, which is supported theoretically by Thm 6.1.
>
> **On Other Comments/Suggestions/References:** Thank you for the detailed suggestions. Due to space restrictions we cannot respond to each in detail. We will add the suggested clarifications and references in revisions. We would like to clarify that in the context of Sec. 2.3, $s$ can be viewed as a “starting time” and $t$ as a time variable at which we evaluate the flow that was started at time $s$. Sec. 2.3 is background on flows and ODEs. Starting with Sec. 3, we substitute a cause variable $x$ to play the mathematical role of time, as stated at the start of Sec. 3.

---

> > ### Comment · Reviewer_H4Vs · 2025-04-02
> >
> > I thank the authors for their response. However, some of my questions have not been addressed.
> >
> > Q1. I understood that the proposed model is more flexible. My question is, if the ground truth is ANM, can we learn the underlying function more accurately using the proposed method? Does the proposed method work better than the HSIC-based criterion?
> >
> > Could you please discuss the optimal transport-based method here?

---

> > > ### Author Response · Authors · 2025-04-04
> > >
> > > We thank the reviewer for their continued engagement. To answer your questions, we have performed additional experiments which have led to additional insights that we think will improve the paper during revisions. Thank you for the questions; we hope our responses have addressed your concerns.
> > >
> > > > If the ground truth is ANM, can we learn the underlying function more accurately using the proposed method?
> > >
> > > Our method is not expected to out-perform fitting an ANM via regression  _in terms of average prediction error_, which is the optimization objective of regression. Nonetheless, our proposed method shows reasonable predictive performance in experiments. Specifically, we considered the data from our original rebuttal experiments (Table 9rbC.1) as well as two extensions. In all cases the ground truth model is an ANM with mean function obtained by randomly sampling weights of a 3-layer MLP. The first extension (_low noise_) reduces the variance of the observational noise from $1$ to $0.2$. The second (_high signal_), in addition to reducing the noise variance, increases the variance of the sampled ground truth mean function weights from $0.2$ to $0.5$.
> > >
> > > For the regression methods, we fit a neural network model to model the mean function in the ANM ($y = m_\theta (x) + \epsilon$) by minimizing MSE, using the same architecture as the ground truth. We did not encounter any optimization issues in any setting, and for regression approaches we observed the MSE loss converging at tolerance 1e-6. See Table H4vs.2 for results. Example plots and fits from the experiment can be found at https://magenta-molli-72.tiiny.site.
> > >
> > > > [If the ground truth is ANM] does the proposed method work better than the HSIC-based criterion [for causal discovery]?
> > >
> > > In short, it depends. We ran an experiment that shows preliminary evidence that our proposed method can be preferable to standard regression (+ HSIC or MSE) when the signal-to-noise ratio is relatively small. Using the same setup as above, we fit regression models and evaluate causal fitness by either the HSIC statistic on the residuals, or by comparing the MSE (we omit the CDT package implementation as it is equivalent to the HSIC approach with a possibly mis-specified mean function). When the data follow a nearly deterministic relationship, standard regression + HSIC/MSE is preferable but our method still works reasonably well (note the joint score can be ill-behaved in this setting). Example plots from the datasets can be found at https://magenta-molli-72.tiiny.site. Table H4Vs.3 shows the performance in three different settings with $N = 4000$.
> > >
> > > > Could you please discuss the optimal transport-based method here?
> > >
> > > Thank you for the reference to Tu, Ruibo, et al. (2022), which we will add to related work in revisions. Although the mathematical definition of velocity overlaps with our method, those authors do not interpret the cause variable as time. Instead, they study a flow from noise to the joint distribution of the observables (in the sense of a continuous normalizing flow) where time acts as an index variable instead of a causal variable, and find conditions on the velocity under which the implied SCM is an ANM to evaluate fitness to the joint distribution. Our framework explicitly treats the cause variable as time, and uses this interpretation to evaluate the fitness of the conditional distribution directly instead of the joint.
> > >
> > > --------------
> > >
> > > **Table H4Vs.2**: Mean square prediction error in causal direction on held-out data. (Training data size N = 4000.)
> > > | | Standard | Low Noise | High Signal |
> > > |---|---|---|---|
> > > | Regression ANM | 0.930 | 0.372 | 0.011 |
> > > | Velocity + Stein ANM | 0.962 | 0.556 | 0.130 |
> > >
> > >
> > >
> > > **Table H4Vs.3**: Causal discovery performance, N = 4000
> > > | Success Rate (AURDC) | Standard | Low Noise | High Signal |
> > > |---|---|---|---|
> > > | Regression ANM+HSIC | 52% (0.58) | 82% (0.93) | 92% (0.99) |
> > > | Regression ANM+MSE | 50% (0.59) | 80% (0.87) | 100% (1.0) |
> > > | Velocity ANM + Stein | 76% (0.87) | 96% (0.99) | 82% (0.93) |

---

### Official Review · Reviewer_sNSP · 2025-03-13

**Overall Recommendation:** 4

**Summary:**

The paper proposes a bivariate causal discovery algorithm utilizing velocity models, viewing structural causal models as dynamical systems where the cause variable acts like time. The approach establishes a relationship between causal velocity and score functions of data distributions, which is exploited for distinguishing cause from effect. The method generalizes beyond traditional additive and location-scale noise models, with promising performance on synthetic and real-world datasets.

**Claims And Evidence:**

- Theoretical claims connecting SCMs are well-supported by the theoretical statements
- Empirical results are convincing
- Proofs (although I have only skimmed over them) seem to be sound
- Authors acknowledge limitations

**Essential References Not Discussed:**

N/A

**Experimental Designs Or Analyses:**

Good and common selection of benchmark datasets (also see Methods And Evaluation Criteria)

**Methods And Evaluation Criteria:**

- Standard data sets and evaluation criteria
- Careful evaluation, although not all baseline values are directly reported (only referenced in the original papers)

**Other Comments Or Suggestions:**

- A brief discussion on computational complexity using the empirical approaches (perhaps in an appendix) would be helpful.
- It is unclear whether the approach could extend beyond the bivariate setting; a small remark on this would help.

**Other Strengths And Weaknesses:**

Strengths:
- Novel approach with an interesting perspective using velocity models
- Generalization of existing models
- Great theoretical justification
- Good figures to follow the ideas

Weaknesses:
- Unclear how to extend it to non-bivariate settings (minor)
- The cause-effect pair performance is not that impressive (related methods are partially better)
- Unclear how well the method performs when assumptions are violated

**Questions For Authors:**

Overall a great paper and well justified approach. Only have a few minor questions/remarks:

- The performance in the cause-effect pair dataset is not that impressive. Are there any insights on the reason why?
- With respect to the previous point, more insights toward violations of assumptions (e.g., non-invertible mechanisms, violation of causal sufficiency etc.) in a more systematic way could be helpful. E.g., the SIM-C dataset contains confounders.
- It seems that the B-QUAD model performs the best on average, is there any insight regarding why?
- An outline of how the proposed approach could be used as functional causal models when modeling a SCM to, e.g., compute Rung 2 and Rung 3 queries in Pearl's ladder of causation can be very insightful (e.g. to reconstruct the noise given a (x,y) pair).

**Relation To Broader Scientific Literature:**

While there are other relevant works, the most important ones were discussed. Overall, a fair selection.

**Theoretical Claims:**

- Correspondence between SCMs and velocity-density pairs, and the connection to score functions, is sound
- The work has an identifiability result (core piece for any causal discovery approach)

---

> ### Author Rebuttal · Authors · 2025-03-31
>
> We thank the reviewer for their careful reading and analysis of our paper. The reviewer lists a few weaknesses and questions that we would like to address.
>
> **Extension to multivariate settings**: Please see our [response to reviewer xsfW](https://openreview.net/forum?id=gV01DWTFTc&noteId=pI8MqzAGXf).
>
> **Performance on cause-effect pair dataset collection**: Indeed, there is room for improvement. There are a number of possible explanations.
>
> - If datasets are generated by a process for which a specific model class (e.g., ANM or LSMN) is well specified, we would expect the specific model-based methods to perform better. Whether that is the case for datasets in the collection is not something we can test.
> - Some of the constituent datasets in the Tuebingen collection have discrete or ordinal data, while our framework requires densities that are continuous on $\mathbb{R}$. We re-ran the experiments after discarding 29 total datasets that have discrete or ordinal data (e.g., “rings” in pair 5 is integer-valued). On this data, our best performing methods (B-QUAD and B-LIN with KDE), obtained 83% (B-QUAD) and 79% (B-LIN) accuracy (up from 59% and 69%, respectively) which is more competitive with other methods. We also re-ran experiments with competing methods in this setting and found that their performance generally did not improve after removal of discrete/ordinal datasets. Please see our [response to reviewer H4Vs, specifically Table H4Vs.1](https://openreview.net/forum?id=gV01DWTFTc&noteId=V3mM0TgYlj) for full results.
>
> **Violation of assumptions**:
> - Non-invertible mechanisms are not an issue for our method. They would be an issue for making unit-level counterfactual inference, but our method only relies on the distributions and their *representations* via functional models (i.e., SCMs), so there is no loss of generality in assuming that the conditional distribution can be represented by a bijective SCM. This is noted after Eq. (1) in the paper, but we will be sure to emphasize the point more clearly in revisions.
> - Confounders: We have not studied this setting and therefore cannot say anything specific, though we consider it important future work. We note that all methods that assume causal sufficiency would likely have issues in the presence of confounders.
>
> **Computational complexity**: We thank the reviewer for pointing this out. We will add a discussion to the appendix. Please see our [response to reviewer xsfW](https://openreview.net/forum?id=gV01DWTFTc&noteId=pI8MqzAGXf) about computational considerations.
>
> **Performance of B-QUAD**: We believe that the B-QUAD model is generally flexible enough to fit most data, while still being simple enough (it only has 9 scalar parameters, see Eqn 67 + 68) to distinguish the causal direction. See Figure 6 of the Appendix for an example of fitting B-QUAD causal curves to real data. Developing a better understanding of this trade-off is an interesting direction for future work.
>
> **Using the velocity parameterization for SCMs**: A great point–we will add a discussion in revision. In short: In the bivariate scalar setting, the velocity parameterization could be used to define novel classes of functional causal models for causal modelling. Since they generate bijective SCMs, they would inherit the counterfactual identifiability results of BCMs [Nasr-Esfahany et al., 2023]. In particular, they can be used to compute queries at all levels of Pearl’s ladder of causation. We can use numerical integration to evaluate the causal curve (Figure 1). Note also that the abduction and prediction steps are especially simple in BCMs, only involving inverting and forward evaluation of the model. In particular for velocity models, this just corresponds to integrating the velocity from an observed condition $x$ to a query condition $x'$, see the discussion above Eqn 2.

---

### Official Review · Reviewer_9rbC · 2025-03-13

**Overall Recommendation:** 3

**Summary:**

This paper delves into how to distinguish between causes and effects in causal relationships through Causal Velocity Models. It proposes a novel framework that treats bivariate Structural Causal Models (SCMs) as dynamical systems and parameterizes these models using causal velocity. The core idea of this method is to infer the causal direction by estimating the score function, without making any assumptions about the noise distribution.

**Claims And Evidence:**

The claims made in the submission are supported by clear and convincing evidence.

**Essential References Not Discussed:**

Most of the essential references have already been cited within this paper.

**Experimental Designs Or Analyses:**

The theoretical analysis of this paper primarily focuses on bivariate causal models. Although the authors mention the potential to extend this method to the multivariate case, specific extension methods and theoretical guarantees have not been discussed in detail.

**Methods And Evaluation Criteria:**

The proposed methods and evaluation criteria make good sense for the problem or application at hand.

**Other Comments Or Suggestions:**

I do not have any other comments or suggestions.

**Other Strengths And Weaknesses:**

Strengths:
1. Unlike traditional causal discovery methods, such as the Additive Noise Model (ANM) or the Location-Scale Noise Model (LSNM), the method proposed in this paper does not require any assumptions about the noise distribution. This makes the method applicable even when the noise distribution is unknown or complex.
2. The method proposed in this paper is not only applicable to traditional ANM and LSNM models but can also be extended to a wider range of model categories. By introducing the concept of Causal Velocity, the authors demonstrate how to parameterize more complex causal mechanisms using basis functions or neural networks.
3. The validity of the method is verified through extensive simulation experiments and benchmark datasets in this paper. The experimental results show that the method can infer causal directions well under various complex data generation mechanisms, especially when existing methods (such as ANM and LSNM) fail, the method proposed in this paper still performs excellently.

Weaknesses:
1. The method proposed in this paper heavily relies on the accurate estimation of the score function. Although the authors have utilized non-parametric estimation methods (such as KDE and Stein's estimator), the accuracy of score estimation may be affected in finite sample scenarios, especially in the tail regions of data distributions.
2. Due to the need for non-parametric estimation of the score function and the use of automatic differentiation to compute derivatives of causal velocity during the optimization process, the computational complexity of this method is relatively high, particularly for high-dimensional data or large-scale datasets.

**Questions For Authors:**

See Weakness.

**Relation To Broader Scientific Literature:**

The key contributions of this paper are related to traditional causal discovery methods, such as the Additive Noise Model (ANM) or the Location-Scale Noise Model (LSNM). Unlike traditional causal discovery methods, the method proposed in this paper does not require any assumptions about the noise distribution. This makes the method applicable even when the noise distribution is unknown or complex.

**Theoretical Claims:**

I have examined all the theoretical proofs to ensure their accuracy.

---

> ### Author Rebuttal · Authors · 2025-03-31
>
> We thank the reviewer for their careful reading and analysis of our paper. The reviewer seems to have appreciated the main benefits of the proposed method (bivariate causal discovery with minimal model assumptions), with some reservations about its reliance on nonparametric score estimation. Our rationale for the reliance on a nonparametric estimate is as follows:
>
> - We view this as an instance of the fundamental statistical trade-off between strength of assumptions and required data/computation. In causality, ANMs sit near the strong assumptions/small sample end of this spectrum. LSNMs already require more data/computation (i.e., two-stage regression) in general. Our method sits further into the weak assumption/more data regime. So do others, such as doubly-robust estimation methods. We would argue that all of these methods are interesting; each is the right tool for some class of problems, with weak assumptions being preferred in settings with limited a priori evidence for favoring specific models.
> - Our approach is to reduce causal discovery to score estimation. We do not claim to introduce a tool that works well on every data set. Our goal is to design a tool that turns reasonable score estimates, where available, into conclusions about cause and effect. This is conceptually similar to methods that reduce causal discovery to independence tests.
> - We find this conceptually appealing since it cleanly separates the causal aspect of the problem (computed once the score estimate is obtained) from the statistical ones. All statistical aspects are, in a sense, encapsulated in the score estimate—the dependence on sample size, for example, only enters through the score estimate.
> - Since score estimation is itself an active research area, we also point out that our method is agnostic to how the score estimate is obtained; it only depends on the quality of the estimate. As new or improved score estimators for a given problem become available, these can simply be plugged into our method.
> - We also note that recent work on causal discovery in ANMs (e.g., NoGAM [Montagna et al., 2023]) also relies on score estimation as a first stage of the discovery method.
> - For computational complexity, please see our [response to reviewer xfsW](https://openreview.net/forum?id=gV01DWTFTc&noteId=pI8MqzAGXf).
>
> ### Discovery with known score
>
> To highlight the separation of causal/statistical components, we performed the following experiment. Using synthetic data generated from an ANM (Gaussian noise; mean function is a 3-layer MLP with random weights and tanh activation) such that the score could be computed analytically/numerically in both directions, we compute the ground truth scores as input to our method and find that sample size plays no role (beyond $n>10$, which is required to evaluate the criterion at a sufficient number of approximation points), and causal direction can be inferred with certainty. We also report the GoF statistic from Eq. 18, showing that it is an order of magnitude lower on average in the causal direction.
>
> **Table 9rbC.1**: Velocity-based causal discovery with known score (results averaged over 100 replications)
>
> |                     | n = 10 | n = 100 | n = 500 | n = 1000 | n = 2500 | n = 4000 |
> |---------------------|--------|---------|---------|----------|----------|----------|
> | GoF Stat Causal     | 0.0459 | 0.0051  | 0.0042  | 0.0041   | 0.0039   | 0.0023   |
> | GoF Stat Anticausal | 0.1067 | 0.0390  | 0.0365  | 0.0365   | 0.0362   | 0.0283   |
> | Success Rate        | 89%    | 100%    | 100%    | 100%     | 100%     | 100%     |
> | AUDRC               | 0.90   | 1.0     | 1.0     | 1.0      | 1.0      | 1.0      |
>
> ### Discovery with estimated score
>
> We repeated the experiment using the same synthetic data as above, using varying sample sizes to estimate the score and subsequently perform velocity-based causal discovery (as in the experiments reported in the paper). We find that as score estimation improves, the success rate of causal discovery also improves.
>
> **Table 9rbC.2**:  Velocity-based causal discovery with score estimated by Stein score/Gaussian kernel (results averaged over 100 replications)
> |                     | n = 10 | n = 100 | n = 500 | n = 1000 | n = 2500 | n = 4000 |
> |---------------------|--------|---------|---------|----------|----------|----------|
> | Score MSE Cause     | 0.0575 | 0.0432  | 0.0169  | 0.0083   | 0.0043   | 0.0033   |
> | Score MSE Effect    | 0.0638 | 0.0417  | 0.0154  | 0.0099   | 0.0049   | 0.0036   |
> | Score MSE Joint     | 0.1436 | 0.1000  | 0.0386  | 0.0219   | 0.0111   | 0.0081   |
> | GoF Stat Causal     | 0.2279 | 0.1165  | 0.0749  | 0.0568   | 0.0429   | 0.0366   |
> | GoF Stat Anticausal | 0.2413 | 0.1235  | 0.0780  | 0.0627   | 0.0495   | 0.0457   |
> | Success Rate        | 49%    | 55%     | 54%     | 58%      | 68%      | 76%      |
> | AUDRC               | 0.54   | 0.59    | 0.59    | 0.69     | 0.77     | 0.87     |

---

### Official Review · Reviewer_xfsW · 2025-03-17

**Overall Recommendation:** 4

**Summary:**

This work studied a class of bijective structural causal models from the perspective of dynamical systems. The identifiablity of the causal model was shown through velocity functions. A loss function is proposed to solve for the velocity function, as well as being used to quantify how well a bijective causal model can be fitted.

Overall, this work is well-written and theoretically grounded.

**Claims And Evidence:**

The claims seem well-supported by theoretical evidence.

**Essential References Not Discussed:**

No

**Experimental Designs Or Analyses:**

The experiment design seems reasonable to me.

**Methods And Evaluation Criteria:**

The designs of the loss and causal discovery procedure are sound.

**Other Comments Or Suggestions:**

No

**Other Strengths And Weaknesses:**

No

**Questions For Authors:**

I have no major concerns about this work. I am curious about the extension of the method to the multivariate case. The current causal discovery procedure does not seem scalable. What are potential ways to improve it in the multivariate setting (if identifiablity can be shown)?

**Relation To Broader Scientific Literature:**

The work established identifiablity for a wider class of bivariate SCMs.

**Theoretical Claims:**

The theoretical results look solid to me, while I did not check correctness.

---

> ### Author Rebuttal · Authors · 2025-03-31
>
> We thank the reviewer for their comments.
>
> **Extension to multivariate settings**: The multivariate case is not a straightforward extension of the bivariate setting. To extend the velocity interpretation we would need to carefully define a multivariate time. We agree that this extension is interesting, but we have not identified a best approach yet and consider this future work.
>
> **Scalability**: The entire procedure, even for $n=5000$, takes less than 3 seconds for KDE estimators, and less than 20 seconds for the Stein estimator on a 2020 M1 Macbook air. The bottleneck is the score estimation, which is asymptotically $O(n^3)$. However, KDE only requires a single matrix multiplication, while Stein only requires several matrix multiplications and inversions, ahead of optimization. One can also use nearest-neighbour based subsampling to reduce the complexity to $O(n \log n)$, which enables scalability to datasets with $n \gg 10000$. Once the score is estimated, optimization of Eq. 18 is a cheap $O(n)$ regression task given the score estimate.
>
> To give this discussion some detail, we report the average computation times/steps for the experiment reported in Table 9rbC.2 of our [response to reviewer 9rbC](https://openreview.net/forum?id=gV01DWTFTc&noteId=yMKZKiiUzH).
>
> **Table xfsW.1**: Computational performance of the proposed method (averaged over 100 replications; velocity optimization convergence tolerance is 1e-6)
>
> |                                                 | n = 10 | n = 100 | n = 500 | n = 1000 | n = 2500 | n = 4000 |
> |-------------------------------------------------|--------|---------|---------|----------|----------|----------|
> | Score estimation time  (Stein w/ Gaussian kernel)                        | 0.027s | 0.034s  | 0.297s  | 0.865s   | 4.600s   | 11.56s   |
> | Velocity optim time (avg)                       | 0.87s  | 0.75s   | 0.63s   | 0.63s    | 0.75s    | 0.79s    |
> | Velocity optimization steps until convergence (causal) | 150.73 | 133.85  | 78.28   | 71.95    | 73.23    | 71.69    |
> | Velocity optimization steps until convergence (anticausal)            | 165.37 | 130.57  | 82.07   | 76.69    | 71.86    | 73.13    |

---

### Decision · Program_Chairs · 2025-05-01

**Decision:**

Accept (poster)

**Comment:**

Reviewers overall agree this submission is a solid theoretical contribution supported by experiment. I therefore recommend acceptance.